# Exceptional freshening and cooling in the eastern subpolar North Atlantic caused by reduced Labrador Sea surface heat loss

Alan D. Fox[1], Patricia Handmann[2], Christina Schmidt[2, 3], Neil Fraser[1], Siren Rühs[2,4], Alejandra Sanchez-Franks[5], Torge Martin[2], Marilena Oltmanns[5], Clare Johnson[1], Willi Rath[2], N. Penny Holliday[5], Arne Biastoch[2,6], Stuart A. Cunningham[1], and Igor Yashayaev[7]

[1]Scottish Association for Marine Science, Oban, UK
[2]GEOMAR Helmholtz Centre for Ocean Research Kiel, Kiel, Germany
[3]now at: Climate Change Research Centre and the Australian Centre for Excellence in Antarctic Science, University of New South Wales, Sydney, NSW, Australia
[4]now at: Institute for Marine and Atmospheric research Utrecht, Utrecht University, Netherlands
[5]National Oceanography Centre, Southampton, UK
[6]Kiel University, Kiel, Germany
[7]Bedford Institute of Oceanography, Fisheries and Oceans Canada, Dartmouth, Nova Scotia, Canada
[*]Corresponding author: alan.fox@sams.ac.uk

**Correspondence:** Alan D. Fox (alan.fox@sams.ac.uk)

**Abstract.** Observations of the eastern subpolar North Atlantic in the 2010s show exceptional freshening and cooling of the upper ocean, peaking in 2016 with the lowest salinities recorded for 120 years. Published theories for the mechanisms driving the freshening include: reduced transport of saltier, warmer surface waters northwards from the subtropics associated with reduced meridional overturning; shifts in the pathways of fresher, cooler surface water from the Labrador Sea driven by changing patterns of wind stress; and the eastward expansion of the subpolar gyre. Using output from a high-resolution hindcast model simulation, we propose that the primary cause of the exceptional freshening and cooling is *reduced surface heat loss in the Labrador Sea*. Tracking virtual fluid particles in the model backwards from the eastern subpolar North Atlantic between 1990 and 2020 shows the major cause of the freshening and cooling to be an increased outflow of relatively fresh and cold surface waters from the Labrador Sea; with a minor contribution from reduced transport of warmer, saltier surface water northward from the subtropics. The cooling, but not the freshening, produced by this changing proportions of waters of subpolar and subtropical origin is mitigated by reduced along-track heat loss to the atmosphere in the North Atlantic Current. We analyse modelled boundary exchanges and water mass transformation in the Labrador Sea to show that since 2000, while inflows of lighter surface waters remain steady, the increasing output of these waters is due to reduced surface heat loss in the Labrador Sea beginning in the early 2000s. Tracking particles further upstream reveals the primary source of the increased volume of lighter water transported out of the Labrador Sea is increased recirculation of water, and therefore longer residence times, in the upper 500–1000 m of the subpolar gyre.

# 1 Introduction

Upper ocean temperature and salinity and their variability in the eastern subpolar North Atlantic are governed by the convergence and divergence of heat and salt and local ocean-atmosphere exchanges. The convergence and divergence are associated with: advection by midlatitude ocean circulation (Piecuch et al., 2017) and the strength of the Atlantic meridional overturning circulation (AMOC) bringing subtropical-origin water northwards (Robson et al., 2016; Foukal and Lozier, 2018; Bryden et al., 2020); northward transport of heat to high-latitudes (Keil et al., 2020); shoaling isotherms and isohalines bringing colder fresher water closer to the surface (Josey et al., 2018); and eastward expansion and contraction of the subpolar gyre (SPG) (Bersch, 2002; Hátún et al., 2005; Bersch et al., 2007; Sarafanov, 2009; Koul et al., 2020), with the accompanying movement of subpolar fronts and changes in the North Atlantic Current (NAC) pathways. The ocean circulation variability is in turn associated with local and remote changes in atmospheric forcing including surface heat fluxes (Josey et al., 2018), shortwave cloud feedbacks (Keil et al., 2020), the leading modes of atmospheric variability in the North Atlantic (Häkkinen et al., 2011; Koul et al., 2020), and wind stress curl patterns (Häkkinen et al., 2011; Chafik et al., 2019; Holliday et al., 2020); as well as upstream freshwater input (Peterson et al., 2006) and internal ocean modes (Johnson et al., 2019).

Multiple observations show exceptional freshening and cooling of the upper 500–1000 m of the eastern subpolar gyre (Josey et al., 2018; Holliday et al., 2020) starting around 2012 and running to the end of the 2010s (Figs. 1 and 2). At its peak, around 2016, this represents the strongest freshening event for 120 years (Holliday et al., 2020). Model and observation based analyses have shown decadal variability in heat and salt content to be primarily driven by variability in northward transport of warm, salty waters from the subtropics (Burkholder and Lozier, 2014; Desbruyères et al., 2015; Robson et al., 2016; Foukal and Lozier, 2018; Desbruyères et al., 2021). This provides decadal-scale preconditioning of the upper ocean by reduced AMOC which, when coupled with intense local winter mixing events bringing colder fresher water to the surface (Josey et al., 2018; Bryden et al., 2020), could produce sufficient cooling and freshening to explain the observed recent exceptional event. Alternatively, increased volumes of cold fresh water from the Labrador Current due to changing pathways of this flow driven by winter wind stress, have also been invoked to explain the exceptional low upper ocean salinity observed (Holliday et al., 2020). This follows more closely ideas of redistribution of heat and salt within the subpolar North Atlantic (SPNA) and variability in the intensity and extent of the SPG driving interannual to decadal variability in the eastern subpolar North Atlantic (Koul et al., 2020; Chafik et al., 2019; Kenigson and Timmermans, 2020). More recent observations of years since 2016 reveal a cooling-to-warming transition induced by increasing transport of warm subtropical waters in the NAC (Desbruyères et al., 2021).

Our aims here are to quantify the relative contributions of annual to decadal variability in ocean circulation, upstream water properties, and local atmosphere-ocean fluxes to the recent exceptional freshening and cooling event of 2012 to 2016 in the eastern subpolar North Atlantic. Using the knowledge gained, we then look at possible mechanisms driving the changes. We use outputs from a hindcast with a high-resolution, eddy-rich ocean model (VIKING20X) (Biastoch et al., 2021), which reproduces the timing and spatial scales and patterns of the exceptional freshening. Using Lagrangian model analysis software (OceanParcels) (Lange and Sebille, 2017; Delandmeter and Van Sebille, 2019), particles, each representing a fixed volume transport, are tracked backwards in time from positions along the upper 500–1000 m of a vertical section spanning the eastern

subpolar North Atlantic. The resulting time series of variability in the volume transports from the water sources, thermohaline properties at source, pathways, transit times and along-track property transformation of the modelled water parcels are used to quantify the contributions to the observed freshening and cooling. We couple this Lagrangian analysis with examination of Eulerian model time series and water mass transformation analysis in the model (Walin, 1982; Tziperman, 1986; Speer and Tziperman, 1992; Nurser et al., 1999) to produce a more complete theory of mechanisms driving the exceptional freshening and cooling.

## 2  Methods

### 2.1  Hydrodynamic model

We make use of the eddy-rich, nested ocean–sea-ice model configuration VIKING20X (Biastoch et al., 2021), based on the Nucleus for European Modelling of the Ocean code (NEMO, version 3.6) (Madec et al., 2017) and the Louvain la Neuve Ice Model (LIM2) (Fichefet and Maqueda, 1997; Goosse and Fichefet, 1999). In the vertical, VIKING20X uses 46 geopotential z-levels with layer thicknesses from $6\,\mathrm{m}$ at the surface gradually increasing to $250\,\mathrm{m}$ in the deepest layers. Bottom topography is represented by partially filled cells allowing for an improved representation of the bathymetry and its slopes (Barnier et al., 2006). In the horizontal, VIKING20X has a tripolar grid with 0.25 degree global resolution, which is refined in the Atlantic Ocean to 0.05 degree, yielding an effective grid spacing of 3–4 km in the subpolar North Atlantic. The run used here, VIKING20X-JRA-short, is an experiment forced from 1980 to 2019 by the JRA55-do forcing (version 1.4) (Tsujino et al., 2018), which is branched from a previous hindcast experiment forced by the CORE dataset (version 2) (Griffies et al., 2009; Large and Yeager, 2009) covering the period 1958 to 2009. Three dimensional model output fields are saved as five day means which are used here for the offline Lagrangian particle tracking and water mass transformation analysis. Hindcasts of the past 50–60 years in this eddy-rich configuration realistically simulate the large-scale horizontal circulation, including the AMOC, the distribution of the mesoscale, overflow and convective processes, and the representation of regional current systems in the North and South Atlantic (Biastoch et al., 2021; Rühs et al., 2021).

### 2.2  Lagrangian particle tracking

Particles are tracked in the VIKING20X model using the Parcels Lagrangian framework v2.2.2 (Lange and Sebille, 2017; Delandmeter and Van Sebille, 2019). About 70,000 particles are released every five days for 30 years, 1990–2019, throughout the top $1000\,\mathrm{m}$ of the OSNAP (Overturning in the Subpolar North Atlantic Programme) line, $\mathrm{OSNAP_E}$ (Fig. 3) east of $37°\,\mathrm{W}$ (Fig. 3). The western extreme was set at $37°\,\mathrm{W}$, and the eastern extreme at the Scottish coast to sample all the northward upper ocean flow in the eastern SPG while excluding the southward upper ocean flow east of Greenland. Each particle is tracked backwards in time for ten years or until it leaves the VIKING20X domain or reaches $20°\,\mathrm{N}$. This is a total of 1.5 billion particle-years. The 1990 to 2019 period was chosen to keep the experiment within the post-1980 hindcast period of the VIKING20X-

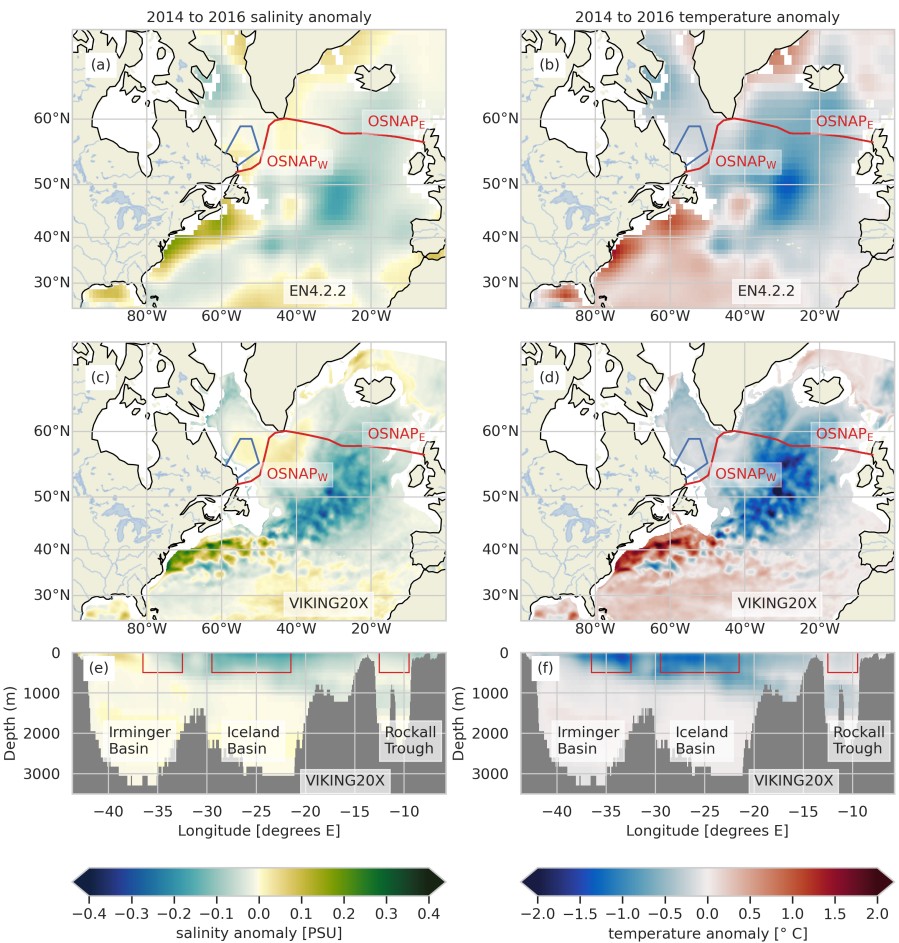

**Figure 1.** The 2014–16 mean **(a, c)** salinity and **(b, d)** temperature anomalies, relative to the 2004–2019 mean, in the upper 500 m from the **(a, b)** EN4 data set and **(c, d)** VIKING20X model. The OSNAP (Overturning in the Subpolar North Atlantic Programme) observational section (**(e, f)** and red lines in **(a)–(d)**), occupied since 2014 and used here to spawn tracked particles in VIKING20X, traverses the fresh patch in the Iceland and Irminger Basins. In 2014–16 there is a widespread region of fresher than average water in the upper 500 m of the eastern Subpolar Gyre, with a 'compensating' region of raised salinity on the North West Atlantic Continental Shelf and Slope region (NWACSS, 40–50° N, 50–70° W, which includes offshore Slope Sea deep water north of the Gulf Stream). The blue polygon in the Labrador Sea outflow region (**(a)–(d)**) shows the region used for comparison with observations of isopycnal depths (Section 6.3). Vertical red lines in **(e,f)** show the position of the time series in Fig. 6.

JRA-short simulation and to compare and contrast the early 1990s and late 2010s subpolar gyre index maxima (Biastoch et al., 2021).

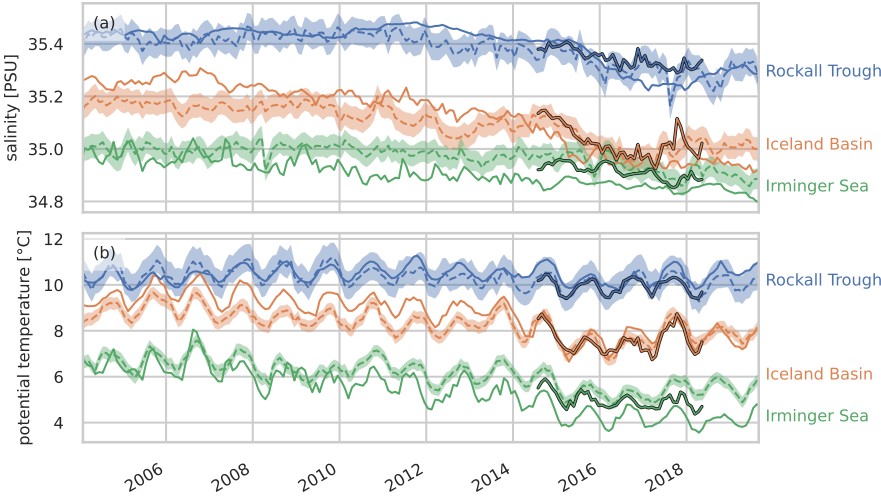

**Figure 2.** Model (plain solid lines) and observational (EN4, dashed lines; OSNAP, bold lines from 2014) time series of **(a)** salinity and **(b)** temperature in three zones – in the Irminger Sea, Iceland Basin and Rockall Trough – along the eastern section of the OSNAP (Overturning in the Subpolar North Atlantic Programme) line, OSNAP$_E$ (see Fig. 1 for position of the zones). All time series are monthly means and averaged over the upper 500 m.

Particles are advected by the VIKING20X 5-day mean, 3-dimensional velocity fields north of 20° N using a 4th order Runge-Kutta time step scheme with a ten minute time step. No additional diffusion is used. Particle position, velocity, temperature, salinity and model mixed layer depth are recorded every five days along-track.

The initial particle distribution along the 2-dimensional OSNAP$_E$ section is random, with local particle density scaled by the magnitude of model velocity normal to OSNAP$_E$ at release time. Each particle therefore represents the same volume transport (here 0.001797 Sv), with positive transport northward and negative southward. Particles are assumed to maintain these along-track transport values throughout the track, as for streamtubes in steady flow (van Sebille et al., 2018). This assumption is only formally valid when using analytical advection methods for steady flows, but Schmidt et al. (2021) have shown that the numerical advection method in Parcels can be reliably used for volume transport estimations.

We attempt to estimate the random errors associated with particle sampling strategy and the effect of unsteady flow on the streamtube assumption. As we increase particle numbers the metrics we discuss will converge towards 'final' values. Using random release positions allows us to split the particle set into a number of subsets (32 here), each of which is still random but with each particle representing a (32 times) larger transport. The spread of the results from these 32 subsets gives us a measure of errors due to the finite sample size. Particle release numbers were set high enough to keep sampling errors small compared to the variability being investigated as is the current standard in following water masses using Lagrangian tools (Schmidt et al., 2021). Possible systematic biases present in all 32 subsets and associated with the model flow fields or the streamtube

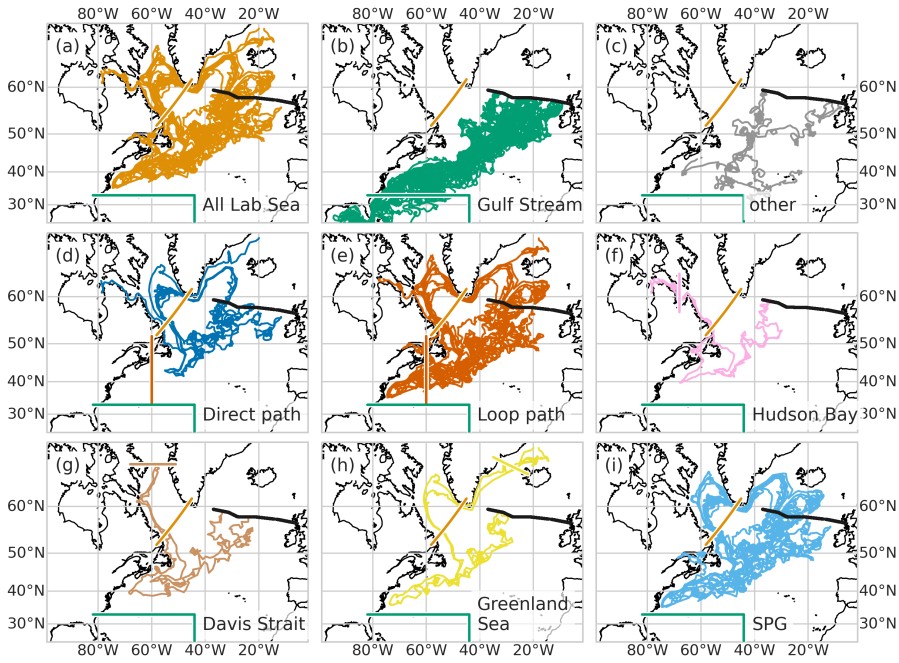

**Figure 3.** Example particle paths showing the identification of particle sources and pathways. The black line is the section of OSNAP$_E$ where particles are released for backward tracking. The coloured straight lines delimit the source regions and pathways: orange – Labrador Sea, green – Gulf Stream, dark orange – the loop pathway, pink – Hudson Bay, brown – Davis Strait, and yellow – Greenland Sea. **(a, b, c)** show the main source regions: **(a)** the Labrador Sea, **(b)** the Gulf Stream, **(c)** no source determined – 'other' source. **(d, e)** show further subsets of **(a)**, highlighting different pathways from the Labrador Sea to OSNAP$_E$: **(d)** the direct path (tracks do not cross the dark orange line at 60° W), **(e)** the loop path (tracks cross and recross the dark orange line). **(f)–(i)** show an alternative subset of pathways of Labrador Sea origin **(a)**, classifying particle tracks by source regions further upstream of Labrador Sea: **(f)** Hudson Bay, **(g)** Davis Strait, **(h)** Greenland Sea and **(i)** particles recirculating in the Subpolar Gyre.

assumption, or errors due to diffusive and eddying processes not being represented in the tracks, will not be quantified by this method.

Particle source regions – Gulf Stream and Labrador Sea – were defined as the region last visited before particles arrived at OSNAP$_E$ (Fig. 3). A small proportion of water (5–10 %, 2–3 Sv) has an undetermined source even after 10 years tracking. These particles with 'other' origin remained circulating within the eastern subpolar gyre for the full ten years of tracking (apart from a very few which came from the Mediterranean Sea). The tracks with Labrador Sea origin were additionally classified by whether they passed through the western Slope Sea region between the Gulf Stream and the shelf slope (which we define as west of 60° W, see New et al. (2021)) – 'loop' path – or took a more direct route – 'direct' path – between leaving the Labrador Sea and arriving at OSNAP$_E$. We further subdivided Labrador Sea origin tracks by source further upstream (Fig. 3f–i): 'Hudson Bay' describes particles which have entered the Labrador Sea from Hudson Bay; 'Davis Strait' describes particles

which entered the Labrador Sea southward through the Davis Strait; 'Greenland Sea' describes particles which have crossed the Greenland-Scotland ridge southward, mostly through the Denmark Strait from the Greenland Sea, before travelling round the Labrador Sea; and 'SPG' describes particles with Labrador Sea origin which have remained within the SPG for a complete circuit.

In the analysis we exclude any particles which pass north of $OSNAP_E$ on their backward tracks between release on $OSNAP_E$ and leaving their source region (see e.g. van Sebille et al., 2018). Here, we mainly present analyses based on a subset of particles which cross $OSNAP_E$ in the surface $500\,\mathrm{m}$ where the freshening is concentrated. We refer to this section as $OSNAP_{E\text{-}37W\text{-}500m}$. Two other sets of particles were explored – crossing $OSNAP_E$ in the top $1000\,\mathrm{m}$ or upper limb of AMOC ($\sigma_0 < 27.62\,\mathrm{kg\,m^{-3}}$). Results from these alternative release configurations are qualitatively the same as the main results, we present some quantitative comparisons between configurations in the text and tables.

Time series are primarily presented plotted against the time the particles cross $OSNAP_E$. It is also useful to present some time series plotted against the time the particles left the major source regions. By grouping particles by their source-leaving time, using the same 5-day intervals as for the particle releases, we obtain the 5-day mean transport leaving the source regions of water which subsequently crosses $OSNAP_{E\text{-}37W\text{-}500m}$. With a ten year tracking duration, these transports measured at source will only be fully 'saturated' between 1990 and 2009, outside this window particle numbers drop off due to the transit times from $OSNAP_E$. To extend this window we use the fractional distribution, $g(t)$, and cumulative distribution, $h(t) = \int_0^t g(t')dt'$ (where $0 < t < 10$ years and $0 < g(t), h(t) < 1$) of transit times to construct a time-dependent normalization factor, $f(d)$, where $d$ is the date in decimal years:

$$
f(d) = \begin{cases} 1 - h(1990 - d) & \text{if } 1980 \leq d < 1990, \\ 1 & \text{if } 1990 \leq d < 2010, \\ 1 - h(d - 2010) & \text{if } 2010 \leq d < 2020. \end{cases} \tag{1}
$$

Particle numbers leaving a source at any time are normalized by dividing by this factor. We cut off the extremes where fewer than $60\,\%$ of the transports would be sampled, giving a window between mid-1987 and mid-2017 (Fig. 4c).

Average Temperature and salinity (TS) properties are calculated both as particles cross $OSNAP_E$ and by averaging T and S from the time each particle leaves the source regions. These averages are performed for the combined sources and for separate sources and pathways. Note that TS averages we present are averaged over the volume being transported, rather than area averages.

## 2.3 Water mass analysis

The rate of formation of a water mass between two isopycnals $M(\rho)$ in some domain can be expressed as

$$
M(\rho) = -\frac{\partial F}{\partial \rho} + \frac{\partial^2 D_{diff}}{\partial \rho^2}, \tag{2}
$$

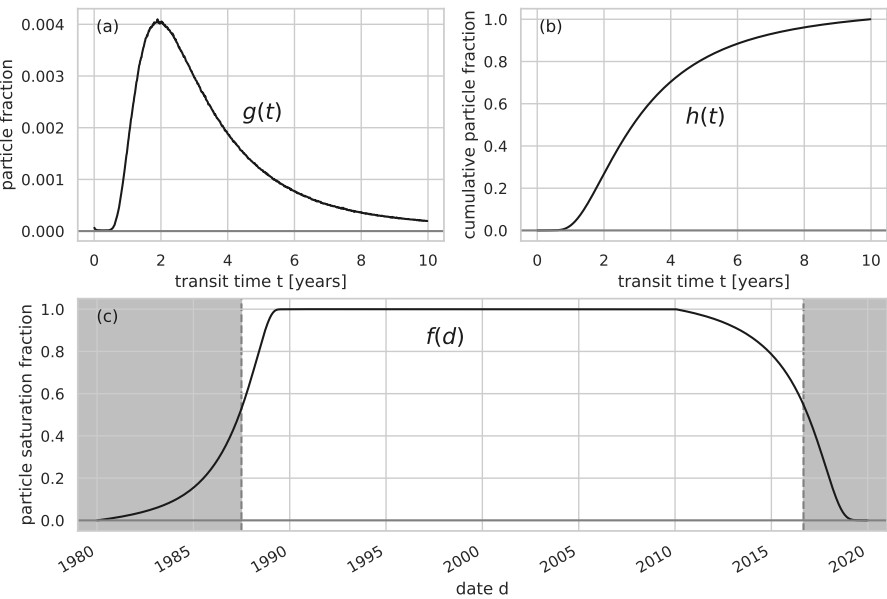

**Figure 4.** Construction of the factor used to normalize the transports at source and select a window of maximum confidence. **(a)** Histogram of transit times between leaving the source region and crossing $OSNAP_E$. **(b)** Cumulative histogram showing the fraction of particles transiting in less than each number of years. **(c)** Particle number normalization factor. Numbers of particles leaving the source are saturated between 1990 and 2010, outside this range numbers fall off at a rate determined by the transit time. Particle numbers leaving a source at any time are normalized by dividing by this factor. We cut off once fewer than 60% of particles are expected to be recorded (shaded zones).

where $F$ represents the the transformation driven by surface fluxes along the surface outcrop of the $\rho$ isopycnal (Speer and Tziperman, 1992), and $D_{diff}$ represents the total diapycnal diffusive density flux across the $\rho$ isopycnal resulting from interior mixing (Walin, 1982; Speer and Tziperman, 1992; Nurser et al., 1999; Fox and Haines, 2003). Here $F$ is given by

$$140 \quad F = \lim_{\Delta\rho\to 0} \frac{1}{\Delta\rho} \int_{outcrop} \mathcal{D}_{in} \, dA, \tag{3}$$

where $\mathcal{D}_{in}$ is the density influx per unit area and $A$ is the surface area outcrop between isopycnals $\rho$ and $\rho + \Delta\rho$. The transformation driven by surface fluxes is divided into heat ($F_H$) and freshwater fluxes ($F_F$) (Tziperman, 1986)

$$F = F_H + F_F, \tag{4}$$

with

$$145 \quad F_H = -\lim_{\Delta\rho\to 0} \frac{1}{\Delta\rho} \int_{outcrop} \frac{\rho\alpha\mathcal{H}}{C_p} \, dA, \tag{5a}$$

$$F_F = \lim_{\Delta\rho\to 0} \frac{1}{\Delta\rho} \int_{outcrop} \rho\beta S\mathcal{Q} \, dA + \int_{surface \ \rho'<\rho} \mathcal{Q} \, dA. \tag{5b}$$

where $\mathcal{H}$ is the downward surface heat flux, $\mathcal{Q}$ is the upward freshwater flux, $\alpha$ and $\beta$ are the thermal expansion and saline contraction coefficients respectively, and $C_p$ is the specific heat capacity.

Integrating Eq. (2) in density, from the lightest $\rho_{min}$ up to $\rho$, we have the volume transformation budget

$$\underbrace{\frac{\partial V(\rho)}{\partial t}}_{\text{volume tendency}} + \underbrace{\Psi(\rho)}_{\text{outflow}} = -\underbrace{F_H}_{\text{surface heat}} - \underbrace{F_F}_{\text{surface freshwater}} + \underbrace{\frac{\partial D_{diff}}{\partial \rho}}_{\text{diffusive outflow}} \tag{6}$$

where $V$ is the total volume of water with $\text{density} < \rho$, and $\Psi$ is the volume flux of water with $\text{density} < \rho$ out of the domain.

Volume tendency, outflow and surface influx terms in Eq. (6) can be calculated from the VIKING20X 5-day mean model outputs, the diffusive flux – which we cannot calculate directly from the standard outputs – can then be backed out as the sum of the other four terms.

## 2.4 Observational data

We use openly available EN4 gridded ocean analyses (EN.4.2.2, bias corrections .g10, downloaded 2021-01-12) (Good et al., 2013; Gouretski and Reseghetti, 2010; Gouretski and Cheng, 2020) for the period with good Argo float coverage (after 2004), OSNAP section gridded data (Lozier et al., 2019; Li et al., 2021) for 2014 to 2018 and RAPID AMOC timeseries (Frajka-Williams et al., 2021) from 2004 to 2020. These data are used with no notable further analysis and details of the collection, quality control and analysis can be found in the relevant publications listed.

For comparisons of model results to observed isopycnal depths in the Labrador Sea we use data from ship, mooring and profiling float observations collected and analysed as part of the Deep-Ocean Observation and Research Synthesis (DOORS), a follow-up to the World Ocean Circulation Experiment (WOCE). All profiles from both vessel surveys and Argo floats operating in the Labrador Sea during 1990-2020 have been quality controlled through semi-automated and visual inspections, calibrated (in case of Argo floats, ship-platform and inter-platform optimizing calibration was developed and used) and validated. The steps of subsequent vertical interpolation and data processing, including evaluation and removal of seasonal cycle, are discussed in http://wwwdev.ncr.dfo-mpo.ca/csas-sccs/Publications/ResDocs-DocRech/2022/2022_039-eng.html. The isopycnal depths were computed for profiles selected covering the Labrador Sea outflow region (blue polygon in Fig 1a-d). Analysis was performed separately for profiles over the Labrador Slope (in water depths of $750\,\mathrm{m}$ to $3275\,\mathrm{m}$) and in deeper water in the central Labrador Sea. The resulting time series of isopycnal depths had the regular seasonal cycle removed, and residuals were low-pass filtered.

## 3 The 2012–2016 eastern subpolar North Atlantic freshening event

First we briefly establish the spatial and temporal characteristics of the exceptional freshening and cooling event in the eastern subpolar North Atlantic beginning around 2012 (Figs. 1 and 2) and show that the VIKING20X model captures enough detail to be a useful tool to evaluate the possible causes (for more detailed description of the observations see (Holliday et al., 2020)). Spatial patterns of salinification/freshening and warming/cooling in the observations and model are very similar (Fig. 1a–d shows 2014 to 2016 anomalies from the 2004 to 2019 means). Both the EN4 observations and the VIKING20X model

show an extensive area of upper ocean freshening and cooling in the eastern subpolar North Atlantic in 2014 to 2016, with strong warming and salinification in the NWACSS (North West Atlantic Continental Shelf and Slope, 40–50° N, 50–70° W, which includes offshore Slope Sea deep water north of the Gulf Stream) region. There is also warming and salinification in the subtropical gyre (south of 35° N), the southern Labrador Sea and north of the Greenland-Scotland ridge in both model and observations. While the large-scale spatial anomaly patterns are similar, VIKING20X anomalies show more small-scale structure than EN4.

Time series of $500\,\mathrm{m}$ depth-mean temperature and salinity in three areas along OSNAP$_\mathrm{E}$ (Rockall Trough, Iceland Basin, and Irminger Sea, Fig. 2) from VIKING20X and EN4 show the fastest freshening and cooling at OSNAP latitudes occurs in the Iceland Basin through 2014 and 2015, with lowest temperatures and salinities recorded in 2016. During the relatively short OSNAP observational period, 2014 to 2018, the model temperature and salinity signals closely track the OSNAP observational data on annual timescales, though not matching all the higher frequency variability. Compared to EN4, in the upper $500\,\mathrm{m}$, VIKING20X has a warm, salty bias in the Iceland Basin between 2004 and 2014, and a cool fresh bias in the Irminger Sea from 2008 to present day.

Vertical sections of model temperature and salinity anomalies (Fig. 1e,f) show the cooling and freshening to be confined to the surface $1000\,\mathrm{m}$ east of the central Irminger Sea, the region occupied by generally northward flowing waters of the upper limb of the AMOC. Deeper waters show warming and salinification. This compares well with changes described in OSNAP section observational data (Fig. 6 of Holliday et al., 2018). While the OSNAP time series is short, it spans the period of fastest upper ocean freshening and cooling.

The evidence presented here shows that VIKING20X reproduces much of the characteristic spatial structure and timing of the exceptional freshening event. As a hindcast, VIKING20X is driven by reanalysis surface forcing but runs freely without data assimilation or relaxation to observations, so the similarity to observations is a product of the external forcing and modelled ocean dynamics. While we do not attempt to quantify the model skill in reproducing the observations, the qualitative comparisons support the idea that we can use the VIKING20X model to help diagnose the causes of the exceptional freshening event. We discuss the possible implications of inaccuracies in the model representation of the freshening event on our conclusions, and caveat those conclusions appropriately, in Section 7.

There are fundamentally only three possible contributions to the exceptional freshening and cooling observed in the eastern subpolar North Atlantic. Firstly, changing proportions of transport coming from the various source regions (including the introduction of new source regions). Each different source region has different average temperature and salinity (TS) properties, so combining them in changing ratios will change TS properties in the eastern subpolar North Atlantic. Secondly, changing TS properties of transports out of the individual source regions. This could alter TS properties in the eastern subpolar North Atlantic even without any change in relative volume transports. And, finally, local processes – surface fluxes and internal mixing – including processes along-track between the source and the eastern subpolar region, causing changing fluxes of freshwater and heat between the upper eastern subpolar North Atlantic waters and the surrounding waters and the atmosphere.

## 4 Lagrangian particle tracking results

### 4.1 Changing volume transports by source and pathway

Annual mean total upper ocean volume transports in VIKING20X across $\text{OSNAP}_{\text{E-37W-500m}}$ (Fig. 5) are high (20 Sv) prior to 1996, reduce by 10–20 % between 2000 and 2009, before returning to 20 Sv after 2016. Isolating these transports by source and pathway shows gradually decreasing volumes of Gulf Stream source waters since 1996. For Labrador Sea source waters the picture is mixed, with the loop path transport declining similarly to the Gulf Stream source transports, while the direct path transports increase rapidly between 2008 and 2016. As a consequence, prior to 1996, water crossing $\text{OSNAP}_{\text{E-37W-500m}}$ had 2–3 times as much Gulf Stream source as Labrador Sea source water, by 2016 the ratio was 0.8–1.2:1 Labrador Sea source to Gulf stream source. Notice differences here with the modelling work of Koul et al. (2020) which, also using particle tracking but in a coarser resolution model, find a maximum of 11 % of near-surface water in the eastern subpolar North Atlantic to be of subpolar (Labrador Sea) origin. In Asbjørnsen et al. (2021), 26 % of the Iceland-Scotland Ridge inflow has a subpolar or Arctic origin (via the Labrador Sea). Here we find never less than 22 %, and up to 45 % of northward transport to be of subpolar origin. These proportions vary depending on the depths and longitude ranges considered. In the 1990s both loop and direct paths from the Labrador Sea paths are equally important with contributions of ∼2.5 Sv each, in the 2010s the Labrador Sea contribution mainly consisted of waters that followed the direct path (∼7.5 Sv), while the loop path contribution becomes small (∼1.0 Sv). The transport volumes are shown (Fig. 5) for the tracks crossing $\text{OSNAP}_\text{E}$ in the upper 500 m, the maximum and minimum transport proportions are the same for tracks crossing in the upper 1000 m or in the upper limb ($\sigma_0 < 27.62\,\text{kg m}^{-3}$) but the total transports increase to 30 Sv and 26 Sv respectively.

Due to the range of particle travel times (Fig. 4a), plotting volume transports against the time at which particles cross $\text{OSNAP}_{\text{E-37W-500m}}$ temporally smooths changes happening at the sources. This introduces the risk of interpreting rapid changes at the sources as gradual changes $\text{OSNAP}_{\text{E-37W-500m}}$ so we also plot volume transports as particles leave the source region. Transports leaving the major sources (Fig. 5b) show similar patterns of variability compared to those recorded at OSNAP, but with source volume transport changes leading OSNAP changes by 3–5 years (corresponding to the most frequent transit times between the source sections and OSNAP, shown in Fig. 4a). In particular transport changes at the sources still appear gradual and maintained over several years, rather than being sudden, step changes. There are signs of an increase in the Gulf Stream and loop path transports leaving the source in 2016, but these increases are yet to be reflected in corresponding increases across $\text{OSNAP}_{\text{E-37W-500m}}$ by the end of 2019.

The general picture over the 30 years of modelled transports across $\text{OSNAP}_{\text{E-37W-500m}}$ is therefore one of gradually reducing volume transports of Gulf Stream origin combined with more rapid increase in total volume transports of Labrador Sea origin after about 2008. This later increase in volume transports of Labrador Sea origin is entirely via the direct path and is new transport rather than a switch in pathways from loop to direct; loop path transports continue slowly declining in parallel with Gulf Stream origin transports. In addition, previously-reported VIKING20X model results Biastoch et al. (2021) show that the denser deep western boundary current (DWBC) transport out of the Labrador Sea decreases after 1996. Together these point to a shift in export volume through the major export routes from the Labrador Sea.

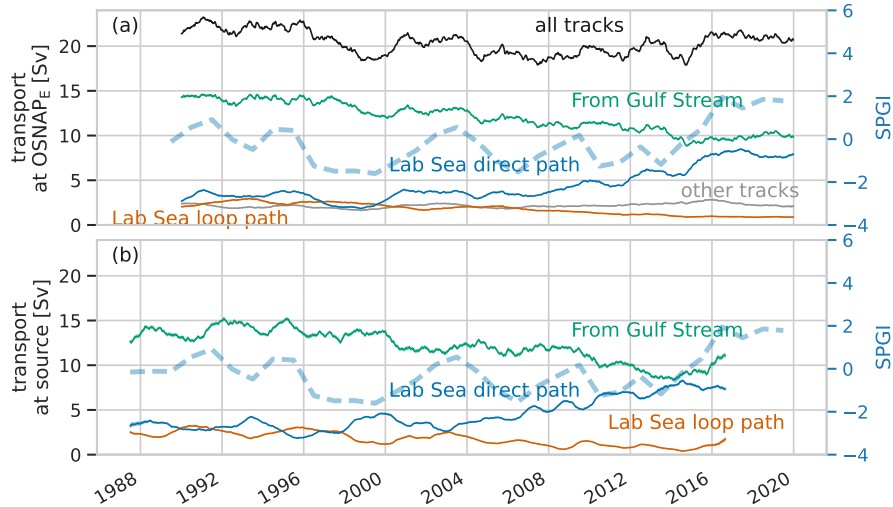

**Figure 5.** Time series of one-year running mean volume transport across OSNAP$_{E-37W-500m}$ by source and pathway plotted against, **(a)**, the time at which water parcels cross OSNAP$_E$ and, **(b)**, the time water parcels leave the source region. The total northward transport (black line) is higher in the 1990s and 2010s and lower in the 2000s. Transport from the Gulf Stream (green) reduces steadily after 1996. From the Labrador Sea, loop path transport (orange) also reduces steadily after 1996 while direct path (blue) increases rapidly after about 2008. Transport with unidentified source after ten years backward tracking is coloured grey. Error bars at $\pm 1.96$ standard deviations around the mean are included but are mostly too narrow to be visible here. The blue dashed line (and right hand y-axis) in **(a)** and **(b)** is the VIKING20X subpolar gyre index (SPGI) from Biastoch et al. (2021).

## 4.2 Temperature and salinity variability, source regions and along-track fluxes

Averaged TS properties at the source, of all particles crossing OSNAP$_{E-37W-500m}$ at time $t$, are calculated from the temperature and salinity of each particle as it leaves its source region. These are volume averages of the water as it leaves the source region. Neither Labrador Sea origin water nor Gulf Stream origin water show freshening or cooling of comparable magnitude to the total freshening and cooling at OSNAP$_{E-37W-500m}$ (Fig. 6). In fact, at their Labrador Sea source, the waters which cross OSNAP$_{E-37W-500m}$ between 1998 and 2012 appear to have become warmer and saltier, with only slight at source freshening and cooling of waters which cross OSNAP$_{E-37W-500m}$ after 2012. Gulf Stream origin water crossing OSNAP$_{E-37W-500m}$ after 2012 showed a small amount of freshening and cooling as it left the source, but as for the Labrador Sea origin this signal is small compared to the total overall freshening and cooling at OSNAP$_{E-37W-500m}$. There is more variability (and more uncertainty) in at source (i.e. 10-years before crossing OSNAP$_{E-37W-500m}$) properties of the 'other' particles, (Fig. 6, grey lines), but these form less than 10 % of the overall transport and only show marked at source cooling and freshening in particles crossing OSNAP$_{E-37W-500m}$ after 2016.

From the transports from the different sources and the mean TS properties at each source we can calculate the 'expected' mean properties of water transported across OSNAP$_{E-37W-500m}$ in the absence of along-track external fluxes (surface fluxes or

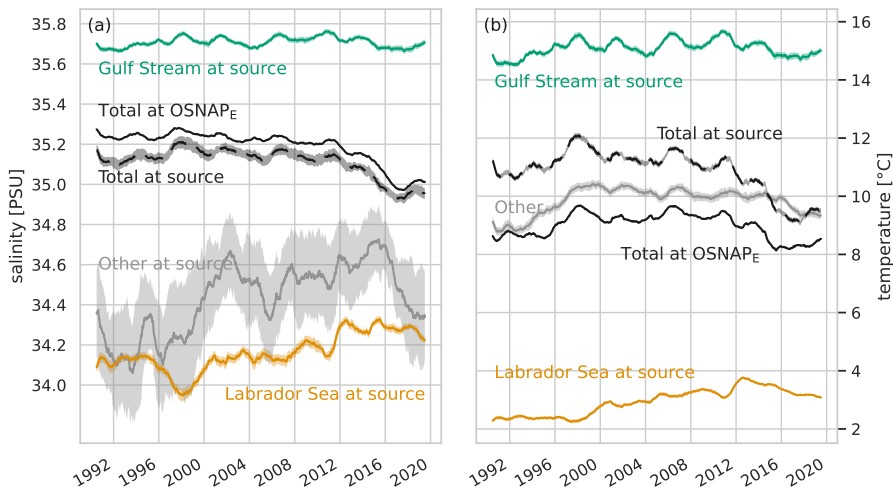

**Figure 6.** Salinity **(a)** and temperature **(b)** time series at the source and at OSNAP$_E$. For salinity, **(a)**, time series of one-year running mean salinity at OSNAP$_{E-37W-500m}$ (solid black lines) and at source (dashed black lines) show the mean freshening at the source to be reflected in overall freshening recorded at OSNAP$_{E-37W-500m}$. This contrasts with temperatures **(b)** where an overall at source cooling (black dashed line) is mitigated by reduced along-track cooling (difference between solid and dashed black lines), resulting in reduced cooling at OSNAP$_{E-37W-500m}$ (black solid line). Dividing the properties at source into the three origins, Gulf Stream (green), Labrador Sea (orange), and other (grey), none shows consistent freshening or cooling. The x-axis is the time water parcels cross OSNAP$_{E-37W-500m}$. Shaded areas show $\pm1.96$ standard deviations between the 32 subsets. Uncertainties are larger for the 'other' source because the sample size is smaller and these particles originate in a wide range of conditions throughout the subpolar North Atlantic south of OSNAP$_E$ rather than in tightly defined source regions.

mixing with surrounding waters) (dashed black lines in Fig. 6). The difference in properties between these 'expected' values
and the recorded properties (solid black lines in Fig. 6) gives the contribution from along-track external fluxes. For salinity these along-track external fluxes act to slightly increase overall mean salinity between source and OSNAP$_{E-37W-500m}$. There is a small consistent trend in the magnitude of this contribution, resulting in weaker along-track salinification after about 1998, hence adding to the net freshening during this period. For temperature, the along-track external fluxes cool the water, as expected with overall heat loss to the atmosphere in the subpolar North Atlantic. In contrast to salinity, the mean along-track external
cooling reduces in time from $2.2\,°\mathrm{C}$ in 1998 to $1.0\,°\mathrm{C}$ in 2017. Note that this reducing trend in along-track externally-driven cooling opposes the observed cooling at OSNAP$_{E-37W-500m}$.

### 4.3  Quantifying the relative contributions to eastern subpolar North Atlantic freshening and cooling

Noting that, by construction, each individual particle transports the same volume, we can write down expressions for the mean temperature and salinity of the volume transported northwards across OSNAP$_E$ at time $t$ in terms of the properties and

transports from each source region and the in-transit change in mean temperature and salinity:

$$\underbrace{\overline{T_o}}_{\text{at OSNAP}_{\text{E}}} = \underbrace{\overline{T_g}\frac{V_g}{V_{all}} + \overline{T_l}\frac{V_l}{V_{all}} + \overline{T_u}\frac{V_u}{V_{all}}}_{\text{at source}} + \underbrace{\overline{\Delta T_t}}_{\text{transit}} \tag{7a}$$

$$\underbrace{\overline{S_o}}_{\text{at OSNAP}_{\text{E}}} = \underbrace{\overline{S_g}\frac{V_g}{V_{all}} + \overline{S_l}\frac{V_l}{V_{all}} + \overline{S_u}\frac{V_u}{V_{all}}}_{\text{at source}} + \underbrace{\overline{\Delta S_t}}_{\text{transit}} \tag{7b}$$

where all variables are functions of the particle release time, $t$. Here $\overline{T_o}, \overline{S_o}$ represent mean T/S properties at OSNAP$_{\text{E-37W-500m}}$; $V_{all}$ is the total volume transport across OSNAP$_{\text{E-37W-500m}}$ at time $t$; $V_g, V_l, V_u$ represent volume transports from Gulf Stream,

Labrador Sea and unidentified 'other' source respectively (and $V_g + V_l + V_u = V_{all}$); $\overline{T_g}, \overline{S_g}, \overline{T_l},$ *etc.* represent mean properties of particles leaving the relevant source; and $\overline{\Delta T_t}, \overline{\Delta S_t}$ denote in-transit changes to mean temperature and salinity.

The dependence of $V_{all}$ on $V_g$, $V_l$ and $V_u$ means we cannot completely separate the contributions to changing temperature and salinity at OSNAP$_{\text{E}}$ from changes to individual properties at, and transports from, the different source regions. But we can estimate each of these contributions by varying the six independent variables one at a time, while holding the others at their

mean values. This will tend to underestimate (overestimate) the influence of changes at one source when transports from the other sources are below (above) their mean values.

Following this process (Fig. 7), qualitatively we find that the dominant driver of variability for both salinity and temperature is the volume transported from the Labrador Sea to OSNAP$_{\text{E}}$, $V_l$. Variability in volume transported from the Gulf Stream $V_g$ plays an additional, but smaller role. Changes in source temperature and salinity show weaker relationships to either overall

mean properties at origin or properties at OSNAP$_{\text{E}}$. Only for temperature, we find a notable influence of in-transit, externally driven changes, $\overline{\Delta T_t}$.

We now quantify the contributions of each of the variables on the RHS (right-hand side) of Eq. 7 to changing properties of water crossing OSNAP$_{\text{E-37W-500m}}$ during the period of most rapid freshening and cooling, between 2011-12 and 2016-17 (Table 1). For salinity, over 60 % of the freshening is due to increasing volumes of water from the Labrador Sea, and 27 %

due to decreasing volumes from the Gulf Stream. The small remainder is a combination of along-track external fluxes and changes in TS properties in the source regions, particularly slight freshening of the Gulf Stream source. For temperature the picture is more complicated, half of the overall mean cooling of water leaving the combined source regions is mitigated along-track by reduced heat loss from the cooler water. The net all-source mean cooling of water leaving the source regions is, as for salinity, driven mostly by increasing volumes of water from the Labrador Sea (over 55 %) and decreasing volume of water from

the Gulf Stream (20 %). As for freshening, a small contribution to the cooling at OSNAP$_{\text{E-37W-500m}}$ stems from temperature changes at the sources, specifically, from a reducing mean temperature of waters leaving the Gulf Stream source. Table 1 also includes results for water crossing OSNAP$_{\text{E}}$ in the upper 1000 m and in the AMOC upper limb ($\sigma_0 < 27.62 \, \text{kg m}^{-3}$). These results are qualitatively the same as for OSNAP$_{\text{E-37W-500m}}$ but with a slightly lower – but always the largest – proportion of the freshening and cooling being due to changing volume transport from the Labrador Sea source region, and higher proportions

due to changing volume transport and source TS from the Gulf Stream source region. These different proportions are due to water of Gulf Stream origin on average occupying deeper levels further east, so as the averaging region goes deeper (to 1000 m)

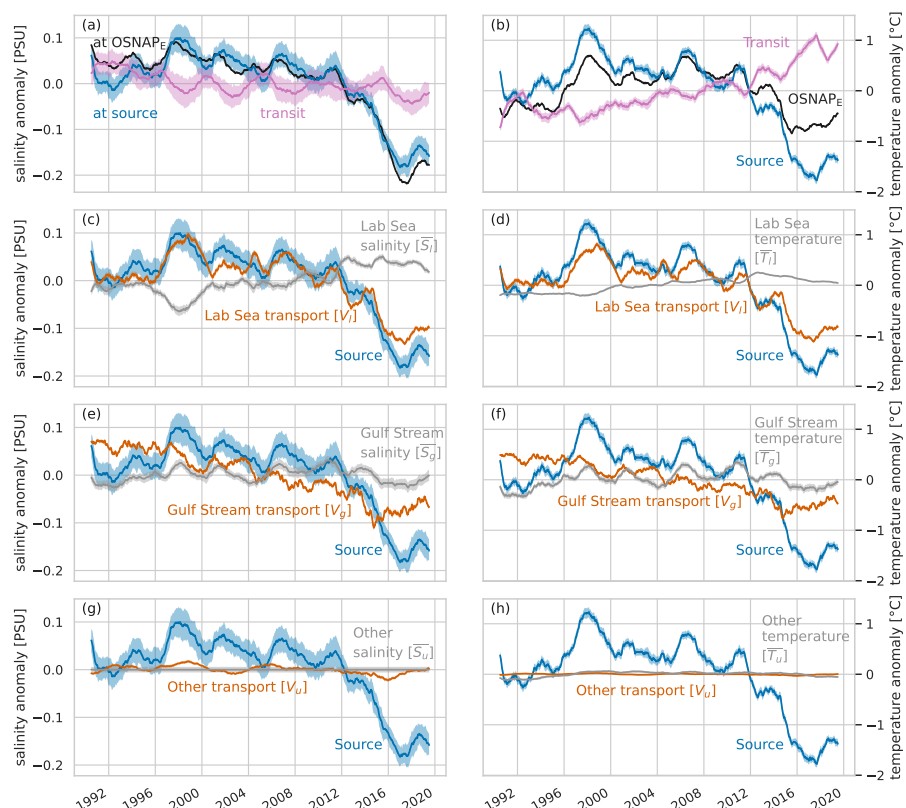

**Figure 7.** Relative contributions of specific causes to the salinity (left, **(a,c,e,g)**) and temperature (right, **(b,d,f,h)**) changes observed at OSNAP$_{\text{E-37W-500m}}$. These are plots of one-year running mean salinity and temperature anomalies. We see that for salinity, **(a)**, the overall freshening at OSNAP$_{\text{E-37W-500m}}$ (black line) closely follows the freshening of the combined sources (blue line) with a small additional contribution from changes during transit (pink). Temperature, **(b)**, shows in-transit changes (pink) opposing source changes (blue). In the panels below, the contributions of six individual processes (orange and grey lines) to the overall source characteristics (blue lines) are estimated, by holding the other five processes at their mean values (see Eq. 7). These six processes are: volume transport variability (orange lines, $V_l$, $V_g$, $V_u$) and salinity/temperature variability (grey lines, $\overline{S_l}$, $\overline{T_l}$, $\overline{S_g}$, $\overline{T_g}$, $\overline{S_u}$, $\overline{T_u}$) for the Labrador Sea source **(c,d)**, Gulf Stream source **(e,f)**, and other source waters **(g,h)**.

and focuses further east (AMOC upper limb) the proportion of Gulf Stream origin water in the mix increases, and with it the relative importance of changes in transports and TS properties of Gulf Stream origin water.

  It is important to note that in the later years modelled, when volume transports of Gulf Stream origin waters are below their 305 30-year mean and those of Labrador Sea origin above their mean, our analysis underestimates the influence of Labrador current origin transports on OSNAP$_{\text{E}}$ properties and overestimates the influence of Gulf Stream origin transports. However, this only further stresses the importance of the described changes in the Labrador Sea for the development of the recent eastern cold and fresh anomalies.

**Table 1.** Contributions to the freshening and cooling of water crossing $OSNAP_E$ in 2016–17 (peak freshening and cooling) relative to 2011–12 (the start of the more rapid freshening and cooling). Contributions are listed by the components described in Section 4.3 and Fig. 7 for water crossing $OSNAP_E$ between $37°$ W and $0°$ in (1) the top 500 m, (2) the top 1000 m, and (3) the upper limb $\sigma_0 < 27.62\,\mathrm{kg\,m^{-3}}$. The largest contribution to both freshening and cooling is made by the changing volume of water of Labrador Sea origin. The single component contributions do not necessarily sum to the total as the component contributions are not independent (see Eq. 7).

| Origin | Component | $37°$ W–$0°$ 0–500 m | | $37°$ W–$0°$ 0–1000 m | | $37°$ W–$0°$ $\sigma_0 < 27.62\,\mathrm{kg\,m^{-3}}$ | |
| | | Salinity change | Temperature change [$°$C] | Salinity change | Temperature change [$°$C] | Salinity change | Temperature change [$°$C] |
|---|---|---|---|---|---|---|---|
| Total | Total | -0.220 | -0.95 | -0.200 | -1.00 | -0.210 | -0.85 |
| Transit | Transit | -0.025 | 1.00 | -0.020 | 0.80 | -0.030 | 1.00 |
| Labrador Sea | Volume | -0.135 | -1.10 | -0.100 | -1.00 | -0.085 | -0.70 |
| | Source TS | 0.015 | -0.05 | 0.000 | 0.00 | 0.000 | -0.05 |
| Gulf Stream | Volume | -0.060 | -0.40 | -0.040 | -0.35 | -0.050 | -0.40 |
| | Source TS | -0.020 | -0.30 | -0.030 | -0.35 | -0.035 | -0.40 |
| Other | Volume | -0.010 | 0.00 | -0.005 | 0.00 | -0.000 | 0.00 |
| | Source TS | 0.000 | 0.00 | 0.000 | -0.05 | 0.000 | 0.00 |

## 4.4 Summary of particle tracking results

We find only small changes in the modelled TS properties of the Gulf Stream and the Labrador Sea origin waters as they leave their respective source regions and also no substantial freshening or cooling through along-track changes. Hence, the exceptional freshening and cooling in the eastern subpolar North Atlantic is driven by the increasing proportion of fresher Labrador Sea origin waters in the mix compared to saltier Gulf Stream origin waters. We have not identified any rapid – annual or intra-annual – changes in source properties, transports or pathways driving the freshening and cooling, but rather pentadal

to decadal, persistent changes beginning in the early 2000s. The increasing volume of Labrador Sea origin water accounts for over 60 % of the freshening, in combination with reduced transport from the Gulf Stream source accounting for about 30 %. For temperature, the shift in relative proportions of water from the two main source regions drives the cooling, but the average cooling of the combined-source waters is mitigated (reduced by about half) by reduced heat loss along-track.

In the following sections, we explore and discuss the possible mechanisms responsible for the modelled changes in the

quantities and proportions of waters transported northwards through the eastern subpolar North Atlantic from the two major sources identified.

## 5 Mechanisms

### 5.1 North Atlantic Current variability and SPG eastward expansion

The distribution of transit times for particles to travel from their source region to $OSNAP_{E-37W-500m}$ (Fig. 4a) show shortest times of under a year and a distribution mode of about 2 years. The distributions are skewed right, with a long tail out to longer transit times. The transit time distributions have a similar shape for both identified source regions and all years, with a steep early rise and long tail at longer transit times, so we can characterise transit time variability by examining variability in the mean. Splitting the section at $21°$ W (Fig. 8), we find contrasting changes over time of mean transit times, speeds and track lengths for particles crossing $OSNAP_{E-37W-500m}$ in the east (Rockall-Hatton Bank and Rockall Trough) and west (Iceland Basin and Irminger Sea).

In the east, over Rockall-Hatton Bank and in the Rockall Trough, northward transports are dominated throughout by waters of Gulf Stream origin. Some increase in transports is seen after 2004 (Fig. 8b) as Gulf Stream source waters are increasingly confined to this eastern section in the later half of the period studied. Transit times, speeds and track lengths to this eastern section of $OSNAP_{E-37W-500m}$ from the individual sources show little consistent change over time (Fig. 8d,f,h) except for a slight slowing of the loop path (though volumes on this path are very small).

In the Iceland Basin and Irminger Sea, across $OSNAP_{E-37W-500m}$ west of $21°$ W, the Gulf Stream influence reduces steadily between 2004 and 2016 and is largely replaced by direct path waters from the Labrador Sea (Fig.8a). Mean transit times in the west, from all sources, increase by around 30 % between 1998 and 2016 (Fig.8c). This is due to a combination of reduced current speeds and increasing track lengths (Fig.8e,g). The increasing track lengths result from eastward shift of the mean $OSNAP_{E-37W-500m}$ crossing longitudes (Fig.8i) and increased along-track eddying (results supported by Eulerian analysis of VIKING20X NAC, not shown). After 2016 transit times from all sources to $OSNAP_{E-37W-500m}$ west of $21°$ W start to decrease, with faster speeds and shorter paths recorded, perhaps associated with a strengthening subpolar gyre from 2013.

These results are consistent with increasing transport in the upper layers of the SPG and eastward movement of the subpolar front between 2006 and 2016, with the reduced volumes of Gulf Stream origin surface waters increasingly confined to the east, passing $OSNAP_{E-37W-500m}$ over Rockall-Hatton Bank and Rockall Trough. The Irminger Sea and Iceland Basin sections of $OSNAP_{E-37W-500m}$ are correspondingly increasingly dominated by waters from the Labrador Sea. The $OSNAP_{E-37W-500m}$ total is dominated by the larger transports west of $21°$ W. Note that our methodology, calculating mean salinity and temperature of volumes transported northwards across the whole upper eastern SPNA (rather than area average salinities and temperatures), precludes pure subpolar gyre eastward expansion (with no new transport) as a cause of the overall cooling and freshening of transport across $OSNAP_{E-37W-500m}$. However, dividing the section zonally suggests that eastward movement of the subpolar front could contribute to redistribution of heat and salt transport along the $OSNAP_E$), causing enhanced local freshening and cooling in the Iceland Basin and reduced freshening and cooling over Rockall-Hatton Bank and in the Rockall Trough. The fractional evolution seen for the eastern section is similar to that seen for the Nordic Seas inflow (Asbjørnsen et al., 2021), consistent with the dominant pathways to the Nordic Seas being on the western flank of Rockall-Hatton Bank and in the Rockall Trough.

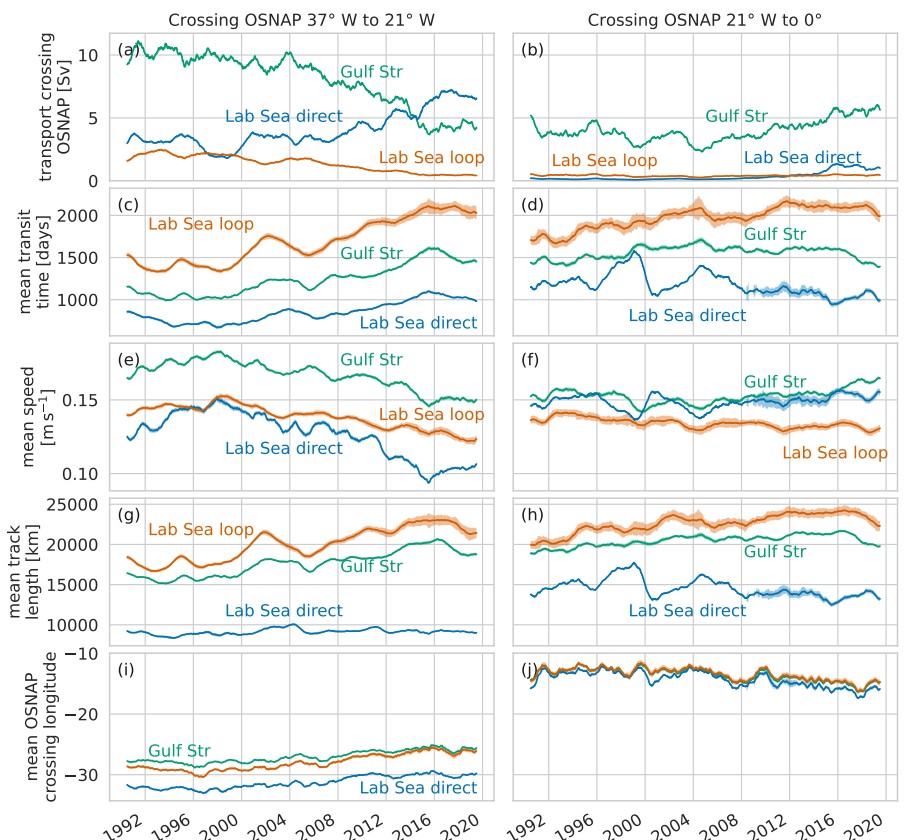

**Figure 8.** Evolution over time of particle transit times. Time series of mean volume transports **(a,b)**, transit times **(c,d)**, speeds **(e,f)**, track lengths **(g,h)** and OSNAP$_{E\text{-}37W\text{-}500m}$ crossing longitude **(i,j)** of particles by source region. Results are divided into OSNAP$_{E\text{-}37W\text{-}500m}$ crossings west (left column) and east (right column) of $21°$ W, the western flank of the Rockall-Hatton Bank. Times on the x-axis are crossing times of OSNAP$_{E\text{-}37W\text{-}500m}$, so values represent weighted averages of conditions along-track in the preceding years. Time series are smoothed with a 1-year rolling mean.

Changing current speeds and pathways in the SPG and NAC have been linked to eastern subpolar North Atlantic freshening (Bersch, 2002; Sarafanov, 2009; Bersch et al., 2007; Hátún et al., 2005; Koul et al., 2020). The proposed mechanism is that eastward expansion and strengthening of the SPG regulates northward transport of warm salty water from the subtropics, modulating the proportion of subpolar and subtropical waters reaching the eastern North Atlantic (Häkkinen et al., 2011). In turn, the strengthening and eastward SPG expansion cause increased freshwater flow into the Nordic Seas (Kenigson and Timmermans, 2020; Asbjørnsen et al., 2021). Using a subpolar gyre index (SPGI) based on the second principal component of sea surface height variability (Koul et al., 2020), Biastoch et al. (2021) show a weak SPG in VIKING20X from 1996–2012, with increasing strength from 2013 (we reproduce this index in Fig. 5). Here we find that, while reduced salinity in the upper eastern subpolar North Atlantic is due to an increasing proportion of subpolar waters, this freshening, starting as early as

2008, appears to lead the increase in SPG strength. We can compare the late 2010s with the previous strong SPG of the early 1990s during which VIKING20X shows an increased SPGI but a much smaller increase in the proportion of subpolar water at OSNAP$_E$ (Fig. 5) and correspondingly smaller cooling and freshening signal in the combined source TS (Fig. 7a,b).

SPG decadal-scale dynamics are complex and still poorly understood. Häkkinen et al. (2011) and Chafik et al. (2019) observe that a strong SPG during periods of eastern subpolar North Atlantic cooling is associated with a strengthened wind-stress curl,
a horizontally expanded gyre, a southeastwards shifted NAC pathway, and reduced advection of warm and saline subtropical waters into the northeast Atlantic. Kenigson and Timmermans (2020) show how the 2014–16 period was characterized by strong positive NAO forcing, strengthened SPG, eastward expansion of the SPG, resulting in freshening in the northeastern North Atlantic Ocean. Alternatively, Robson et al. (2016) propose that cooling of the upper eastern subpolar North Atlantic is due to reduced transport of warm water from the south associated with decreased AMOC, which in turn is linked to record low
densities (lighter water) in the deep Labrador Sea.

The VIKING20X model results which we present here suggest a decoupling of the eastward expansion of the SPG from the strength (as measured by an SSH-based SPGI) with eastward expansion beginning during a period of weaker SPG. The cooling we observe in transports at OSNAP$_{E-37W-500m}$ beginning before 2012 is mainly linked to an increasing proportion of subpolar water (Fig. 7), which is comprised of increasing volumes of water which left the Labrador Sea from around 2008
(Fig. 5). While we cannot discount SPG variability from contributing to the exceptional cooling, the trends in upper-layer volume transports, which drive the freshening and cooling in the eastern subpolar North Atlantic, begin before 2008 in a period of weak, and possibly still weakening (Biastoch et al., 2021), SPG.

## 5.2 Subtropical origin water and meridional overturning

There is currently no consensus on whether the AMOC has declined since the 1990s (see for example Jackson et al., 2022;
Latif et al., 2022, for summaries of the debate). Models and proxies suggest a decline (e.g., Rahmstorf et al., 2015; Caesar et al., 2018, 2021) while observation-based analyses give a mixed picture, some finding no significant AMOC decline (Fu et al., 2020; Worthington et al., 2021; Caínzos et al., 2022), the RAPID Meridional Overturning Circulation and Heat-flux Array (RAPID-MOCHA) (Cunningham et al., 2007) finding reduced AMOC after 2009 (Bryden et al., 2020), and application of the Bernoulli inverse to hydrography (Fraser and Cunningham, 2021) showing weakening AMOC during our experiment
period.

We here show results at 29° N (Fig. 9), a latitude close to the RAPID observation line and to our Gulf Stream source definition latitude. We choose 29° N, near the meridional centre of the subtropical gyre, because of the simple configuration of the large-scale flow at this latitude. The western boundary current (Gulf Stream), containing almost all the northward warm-water transport, sits entirely on the 1000 m deep shelf; the balancing return southward flow is made up of the deep-western boundary
current – the lower limb of AMOC – along the slope at the shelf edge, and the broad southward flow in the upper 1000 m of the interior and east of the subtropical gyre – which we designate subtropical gyre recirculation. At 29° N in VIKING20X-JRA-short, the Gulf Stream flow weakens between 1990 and 2020, this is accompanied by reduced AMOC and therefore reduced lower limb return flow. The remaining southward return flow, the subtropical gyre recirculation, shows no overall trend. So the

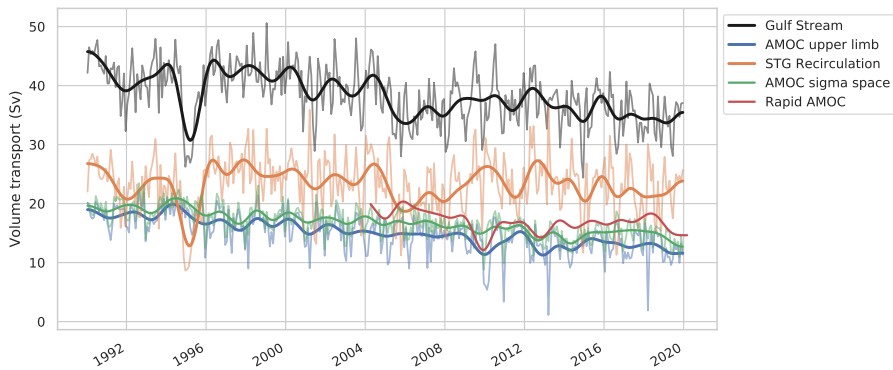

**Figure 9.** Northward transports and AMOC strength at $29°$ N in VIKING20X. Pale lines are monthly mean data, darker lines are low-pass filtered using a Butterworth filter and 18 month cut-off. The total transport in the Gulf Stream (black line) reduces throughout the experiment duration, while the volume recirculating in the subtropical gyre above $1000$ m (light orange line) shows a smaller trend or remains largely constant. This implies weakened AMOC and reduced northward flow in the AMOC upper limb (blue, AMOC in z-space). At $29°$ N, the AMOC in $\sigma$-space (green) is very similar to $AMOC_z$ and also reduces consistently. The dark orange line is AMOC in z-space from RAPID observations, lowpass filtered at 18 months, for comparison. The reducing AMOC strength (blue and green) closely reflects the reducing contribution of Gulf Stream origin transport at $OSNAP_E$ found from the particle tracking.

Gulf Stream weakening in VIKING20X is associated with reduced flow northwards from the subtropical towards the subpolar
North Atlantic in the upper layers and corresponding reduced deep return flow – that is reduced AMOC. This weakening AMOC, by about $30$ %, is associated with the reduction in modelled transport of Gulf Stream origin to $OSNAP_E$ between the 1990s and 2016 (Fig. 5) diagnosed by the Lagrangian tracking.

Previous longer-term, lower-resolution modelling work (Koul et al., 2020) linked periods of higher subpolar gyre index to weaker flow from the subtropical gyre to the NAC, suggesting a possible 'blocking' mechanism. But, the AMOC weakening
in VIKING20X between 1990 and 2020 appears to be independent of the subpolar gyre strength. In this period, the AMOC weakening contrasts with a high (1990s) to low (2000s) and back to high (late 2010s) cycle in the subpolar gyre index.

Examination of the decadal scale drivers of the AMOC, in either the real ocean or in VIKING20X is beyond the scope of the current work, but we refer the reader to Biastoch et al. (2021) for a detailed analysis. They find outflow of deep water from the Labrador Sea at $53°$ N to be a good indicator of the subpolar AMOC trend and also a strong dependence on the choice of
freshwater forcing in the model. The VIKING20X-JRA-short run used here shows a stronger decline in the AMOC than either observations or other VIKING20X realisations since 1990, but all consistently suggest a decline for this period.

Comparing the VIKING20X AMOC time series since 2005 with the observational time series from RAPID (see e.g. Bryden et al., 2020), in VIKING20X the AMOC appears $3$–$4$ Sv too weak but the model reproduces the observed overturning minimum in 2009 and the reduction in overturning by $2$–$3$ Sv from before 2009 to after 2009. The 2009 minimum is suggested by Bryden
et al. (2020) as the start of a period of reduced northward transport of heat and salt from the subtropics to the subpolar gyre. The similarities in AMOC at the RAPID array between model and observations since 2005 give us confidence that we could be, at

least qualitatively, reproducing the contribution of changes in overturning circulation to reduced eastern North Atlantic salinity and temperature. In the first half of our time series, before the start of the RAPID observations, VIKING20X shows declining AMOC with consistent weakening at latitudes throughout the North Atlantic (Biastoch et al., 2021). Compared to observation-based estimates, this AMOC decline between 1990 and 2005 in VIKING20X may be an overestimate, which could in turn lead to us overestimating the role of the AMOC weakening in the observed eastern subpolar gyre freshening and cooling.

In summary, in Section 4.3 we found that reduced transport from the Gulf Stream source accounted for about 30 % of the cooling and freshening observed in the model at OSNAP$_{E-37W-500m}$. Examining this in more detail shows a longer-term gradual decline in subtropical water transported across OSNAP$_{E-37W-500m}$ between the mid-1990s and 2016, contrasting with the accelerating freshening and cooling from 2011–2016. This decline is associated with AMOC weakening in VIKING20X. The reduced transports of Gulf Stream origin will amplify TS variability in the eastern subpolar North Atlantic due to variability in subpolar gyre transport from the Labrador Sea origin (Eq. 7).

## 5.3 Subpolar and Arctic origin water

### 5.3.1 Pathways from the Labrador Sea

Upper water from the Labrador Sea reaches the eastern subpolar North Atlantic along one of two dominant pathways (Holliday et al., 2020; New et al., 2021): a direct pathway, and a loop pathway which follows the coast southwestwards through the Slope Sea north of the Gulf Stream before reversing, joining the Gulf Stream origin waters travelling to the northeast (Fig. 3d,e). A change in the dominance of the direct pathway for cold fresh Arctic-origin water from the Labrador Current has been offered as a cause of the exceptional freshening event (Holliday et al., 2020).

Examining the changing volume transports along these two pathways between the Labrador Sea (Fig. 5) and OSNAP$_{E-37W-500m}$ shows the ratio of the direct to loop source volume changes from close to 1:1 before 2000, to 8:1 after 2016. Prior to 2006, dominance switches repeatedly between the two pathways with little change in total transport (Fig. 5b). This early period supports the ideas of changing pathway dominance of Holliday et al. (2020), but after 2006 the net increase in volumes leaving the Labrador Sea towards OSNAP$_{E-37W-500m}$ in VIKING20X is entirely along the direct path.

It is this more recent increase in total volume of Labrador Sea source water, rather than by the details of pathways taken, which we showed in Section 4 is responsible for the major part of 2012–2016 freshening and cooling in the eastern subpolar North Atlantic. In the following sections we explore possible mechanisms driving this increased total transport from the Labrador Sea.

### 5.3.2 Source regions upstream of the Labrador Sea outflow

Increased transport from the Labrador Sea must be associated with changes in transport from source regions upstream of the Labrador Sea outflow. Tracking the water of Labrador Sea origin back in time, further upstream (Fig. 10), we can distinguish between four upstream source regions: Hudson Bay; the Davis Strait; from the Greenland Sea, southward across the Greenland–Scotland ridge; and SPG water (particles circulating in the SPG, and having previously passed northwards across OSNAP$_E$).

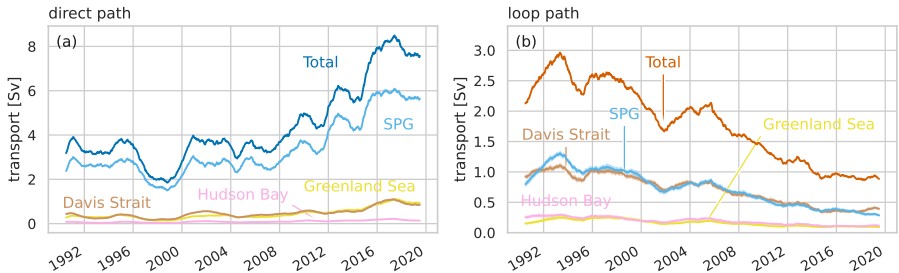

**Figure 10.** Source regions upstream of the Labrador Sea outflow. Quantifying contributions from sources upstream of the Labrador Sea outflow for the direct path ((**a**) and the loop path (**b**)). Upper layer inflow to the Labrador Sea comes from Hudson Bay (pink), Davis Strait (brown) and in the West Greenland Current. The West Greenland Current is further divided here into water from the Greenland Sea (yellow) and recirculating SPG origin water (light blue). There is little change in the proportions of water from different upstream sources forming the different paths (loop or direct). Overall, the increased contribution of the direct path (**a**) is reflected in an increase in the total volume of recirculating SPG transport and a small increase in the total contribution from the Greenland Sea. Particles from Hudson Bay, Davis Strait, and the Greenland Sea have come directly from those regions, SPG origin waters are circulating in the subpolar gyre and have previously crossed northwards across OSNAP$_E$.

The overall increase in volume of Labrador Sea origin water (4 Sv) is primarily originating from an increase in circulating
SPG water (3 Sv) with a smaller contribution from increases in waters of Greenland Sea and Davis Strait origin (1 Sv) (Fig. 10). While volumes taking the direct path increase and those taking the loop path decrease, we find little variability, and no trend, in the percentage of waters from the four upstream source regions forming either path. The direct path is about 80 % SPG origin water, 10 % Davis Strait origin and 10 % Greenland Sea origin; the loop path is 40 % SPG origin, 40 % Davis Strait origin, 10 % Greenland Sea origin and 10 % Hudson Bay origin. All the flows we are examining here are in the light, upper layer
waters which cross OSNAP$_E$ in the surface 500 m.

There is little change in total volume contribution from Hudson Bay or from Davis Strait upstream source regions, although the dominant pathway taken by the Davis Strait origin water switches from loop to direct. This is exactly the switch of dominance described by Holliday et al. (2020).

The small increase in contribution of upper waters from the Greenland Sea is interesting, perhaps suggesting increased
Greenland meltwater flowing southward in the East Greenland Current. The fate of this fresh water in regions of deep water formation has led to speculations that the accelerating melting of the Greenland Ice Sheet could stratify the subpolar gyre, change the spatial deep convection patterns, and slow or stop the AMOC (see for example Böning et al., 2016; Foukal et al., 2020; Rühs et al., 2021). This increase accounts for <20 % of the increase in transport from the Labrador Sea to OSNAP$_{E-37W-500m}$.

The largest part ∼75 %, 3 Sv, of the increased transport from the Labrador Sea is due to increased volumes recirculating
within the upper North Atlantic subpolar gyre. This recirculation doubles, from 3 to 6 Sv, between the early 2000s and 2016, representing an increase from ∼15 % to ∼30 % of the northward transport across OSNAP$_{E-37W-500m}$. It is this recirculation which is driving the increase in transport from the Labrador Sea to OSNAP$_{E-37W-500m}$, and in turn the freshening and cooling.

There are various possible mechanisms behind this increased contribution from water recirculating within the near-surface SPG. One is an accelerating SPG, but in Section 5.1 we showed the timing of the recirculation increase (from 2008) and the major SPG acceleration (after 2013) suggest this is an unlikely cause. To leading order, water transported northwards in the upper layers of the eastern subpolar North Atlantic generally does one of three things: crosses the GSR northwards, is made denser in the Nordic and Arctic seas, and returns in deeper overflows; or is made denser within the SPG, sinking to intermediate and deep layers; or continues to circulates within the SPG. To increase the proportion recirculating, one of the other pathways needs to weaken, suggesting reduced densification processes in either the subpolar gyre or Nordic and Arctic seas. We return to these ideas in Section 6).

### 5.3.3  Labrador Sea outflow

A VIKING20X mean velocity section across the shelf and shelf break at the western end of the $OSNAP_W$ line spans the upper layer outflow, the Labrador Current, the source region of all the modelled transport from the Labrador Sea to $OSNAP_{E-37W-500m}$ (Fig. 11a). This shows upper layer outflow to be concentrated in two main cores: one close to the coast – Labrador Coastal Current containing Hudson outflow and water from the Davis Strait; and the other (larger and stronger) over the shelf break and slope. Most of the water exiting the Labrador Sea with a density ($\sigma_0$) lighter than 26.65–26.70 $\mathrm{kg\,m^{-3}}$ is transported round to $OSNAP_{E-37W-500m}$ in the surface 500 m. Pentadal mean velocity anomalies (Fig. 11b–g) across this Labrador Sea outflow section show no coherent overall increase in outflow velocities (negative, blue, in Fig. 11(b)-(g)).

The offshore current has previously been separated into distinct shallower inshore (upper 200 m inshore of the 600 m isobath) "LC-Arctic" and deeper offshore "LC-Atlantic" components (Florindo-López et al., 2020; Holliday et al., 2020; New et al., 2021). While the definitions of these outflow components do not map precisely onto our upstream origins (Section 5.3.2), the LC-Arctic component is water of Arctic origin and is primarily made up of waters we describe as Davis Strait and Greenland Sea origin, and the LC-Atlantic outflow corresponds closely to our recirculating SPG origin water. The LC-Arctic and LC-Atlantic components are not easily separable in the VIKING20X mean outflows at $OSNAP_W$ (Fig. 11a), though some variability in the LC-Arctic component is visible near-surface at about 360 km offshore in the pentadal velocity anomalies (Fig. 11b-g). The VIKING20X anomalies in the LC-Arctic outflow are consistent with Florindo-López et al. (2020), showing higher outflow in the early 1990s and late 2010s and reduced outflow between. Note that the total LC-Arctic outflow is typically less than 2 Sv (New et al., 2021), so the increased LC-Arctic outflow after 2005 forms a minor component of the total outflow increase seen in VIKING20X, which is dominated by the offshore LC-Atlantic, SPG water.

Pentadal mean density fields show deepening of all isopycnals between 27.60 $\mathrm{kg\,m^{-3}}$ and 27.75 $\mathrm{kg\,m^{-3}}$ between the 1990–94 and 2010–2014 pentads (Fig. 11b—g), followed by a slight recovery to shallower depths by 2015–2019. It is the increase in cross-sectional area occupied by these outflowing light waters (27.60 $\mathrm{kg\,m^{-3}} < \sigma_0 < 27.75\,\mathrm{kg\,m^{-3}}$), rather than increased flow speeds, which produces the increased transports of light upper waters out of the Labrador Sea.

Note that there is no obvious increase or decrease in cross-sectional area occupied by waters with $\sigma_0 < 27.30\,\mathrm{kg\,m^{-3}}$. These surface waters on the shelf form the Labrador Coastal Current which, combined with some inshore near-surface water from the shelf break, mainly follow the loop path. Labrador Current outflow along the loop path has been found to be related to Gulf

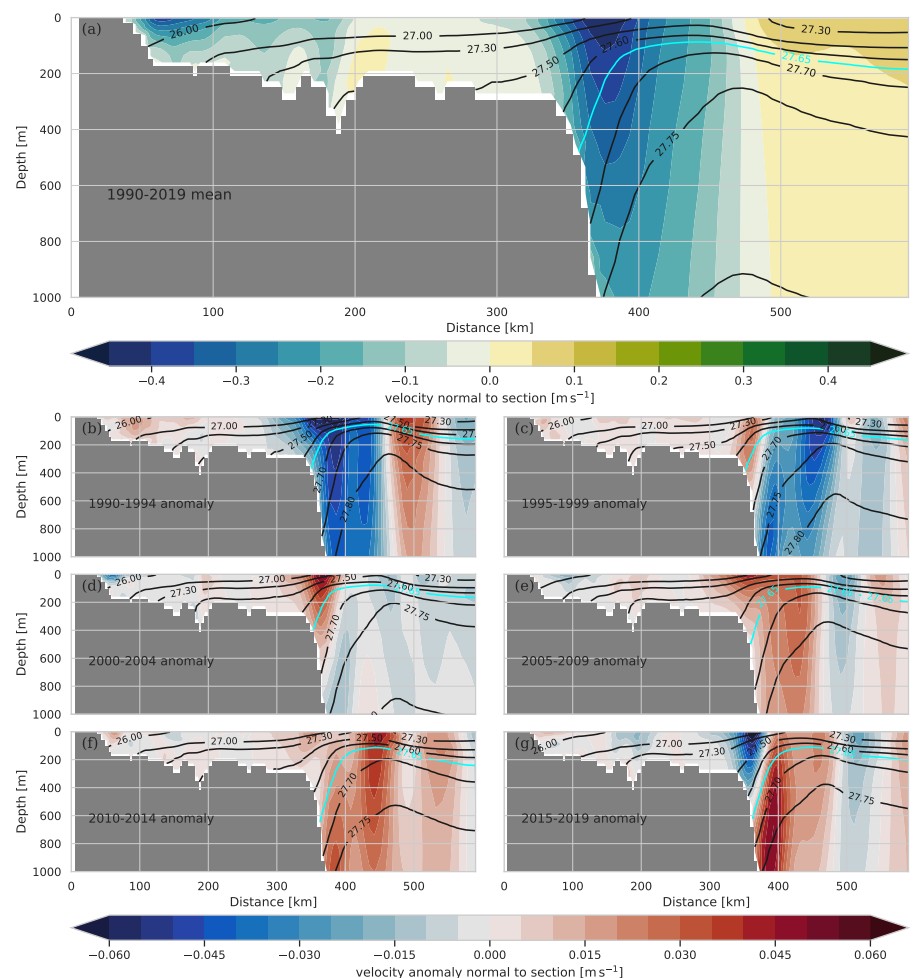

**Figure 11.** Vertical sections of Labrador Sea upper layer outflow 1990–2019 mean velocity **(a)** and pentadal velocity anomalies from the mean **(b)–(g)**, with period-mean density contours superimposed, at the western end of the OSNAP line. Velocities, and velocity anomalies, are shown as positive into the Labrador Sea. Lighter waters (<27.65 kg m$^{-3}$) leaving the Labrador Sea are mostly transported to OSNAP$_E$ in the subpolar gyre, this transport is dominated by strong currents at the shelf break. Outward currents in the lighter density classes are weaker (positive – red – anomalies) than the 30-year average between 2000 and 2015 (**(d)-(f)**). The increasing contribution of Labrador Sea origin waters to eastern SPG transports in the 2010s is associated with deepening of these lighter layers rather than increasing currents, notice the highlighted 27.65 kg m$^{-3}$ isopycnal in panels **(b)–(g)**.

Stream transport and AMOC strength (New et al., 2021; Sanchez-Franks et al., 2016) possibly via wind stress in the subpolar gyre.

Qualitatively, we see the potential for increasing volume transport from the Labrador Sea to the upper eastern subpolar North Atlantic to be due to a deepening of the surface layer of lighter water in the Labrador Sea rather than faster currents. We now

explore that idea further, examining the increasing outflow of lighter waters, using a water mass transformation budget in the Labrador Sea Walin (1982); Speer and Tziperman (1992).

## 6 Labrador Sea water mass analysis

### 6.1 Full water mass balance

To first order, the Labrador Sea is fed by water from the West Greenland Current, through the Davis Strait, from Hudson Bay together with river, meltwater and precipitation inflows; this water is transformed to denser water within the Labrador Sea by winter cooling and ice formation before exiting southwards via the Labrador Current in the upper ocean and the DWBC below.

The mean water mass transformations (Walin, 1982; Speer and Tziperman, 1992) on the ocean model volume bounded by OSNAP$_W$, Davis Strait at $67°$ N and the exit of Hudson Bay at $68°$ W (Fig. 12) confirm this balance. On average, net boundary

inflow (inflow - outflow) acts to increase the volume of water lighter than $\sigma_0 = 27.70\,\mathrm{kg\,m^{-3}}$, and reduce the volume of denser waters. This net inflow is primarily balanced by the surface heat loss, cooling the lighter inflow waters and hence transforming them to higher densities before they flow out of the Labrador Sea. The contribution to transformation by surface freshwater fluxes (including river inflow, ice melt/formation and precipitation-evaporation) is much smaller than the cooling. There is also a small change in the relative volumes of water within the Labrador Sea, with increasing volumes of waters in most density

classes balanced by decreasing volumes of the densest waters.

### 6.2 Changing water mass balance in the upper Labrador Sea

From water mass transformation theory we know that the net rate of inflow water $\rho < \rho_0$ is balanced by the rate of change of volume with $\rho < \rho_0$ and the rate of transformation of water across the $\rho = \rho_0$ isopycnal. This transformation rate is a function of the surface heat and freshwater flux at $\rho = \rho_0$ isopycnal outcrop, and the transformation by interior mixing across the $\rho = \rho_0$

surface. Figure 13 shows how this balance evolves over time for lighter waters of $\sigma_0 < 27.65\,\mathrm{kg\,m^{-3}}$. We also further split the net inflow into 'inflow' and 'outflow' components by assuming the inflows to be through Davis Strait, Hudson Bay, and across OSNAP$_W$ north of $57°$ N (mostly the west Greenland Current), and all outflow to occur across OSNAP$_W$ south of $57°$ N. It is the variation in time of this 'outflow' in the southern half of OSNAP$_W$ we are interested in.

The $\sigma_0 = 27.65\,\mathrm{kg\,m^{-3}}$ density level is chosen because, in VIKING20X, it represents the approximate density lighter than

which most water exiting the Labrador Sea is transported across to OSNAP$_E$ in the upper layer. Coincidentally, it is close to the division between the upper and lower limbs of the meridional overturning at the OSNAP line. We will use 'upper limb water' and 'lower limb water' as shorthand for water lighter and denser than $\sigma_0 = 27.65\,\mathrm{kg\,m^{-3}}$, respectively. The precise choice of density level, between $\sigma_0 = 27.65\,\mathrm{kg\,m^{-3}}$ and $\sigma_0 = 27.70\,\mathrm{kg\,m^{-3}}$, does not affect the conclusions. The density $\sigma_0 = 27.70\,\mathrm{kg\,m^{-3}}$ is often considered to be the lighter density limit of the deeper LSW intermediate water masses in the

North Atlantic.

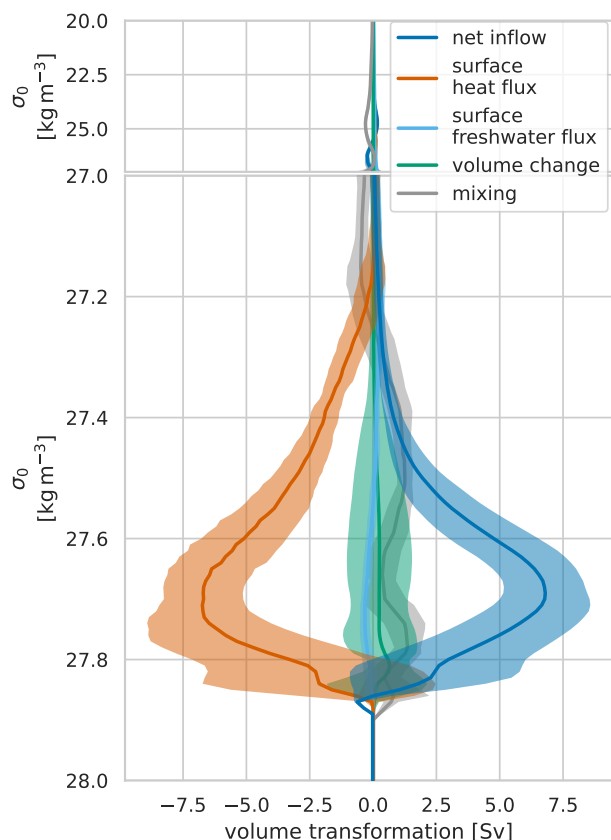

**Figure 12.** VIKING20X Labrador Sea mean water mass balance. Positive values of volume transformation are transformations across the isopycnal towards lighter densities. Net production of water of a particular density is therefore shown as negative gradients and net removal as positive gradients (i.e. surface heat flux acts to remove lighter waters by cooling – positive gradients – producing denser waters – negative gradients). The primary balance is between net inflow of lighter water and conversion to denser water by surface heat flux. Solid lines are average annual mean transformations, 1990-2019, shaded area shows $\pm 1$ standard deviation in the annual mean.

On interannual and longer timescales, after 2000, the modelled inflow of upper limb water to the Labrador Sea shows some variability but no trend. The outflow of these waters after 2000, while always being less than the inflow, has an increasing trend (Fig. 13a) closely resembling the timing of, but of slightly larger amplitude than, the increase of the Labrador Sea origin transport observed in the particle tracking. The amplitude difference is because there is not a 1:1 correspondence between

Labrador Sea outflow $\sigma_0 < 27.65 \, \text{kg m}^{-3}$ and water parcels tracked to OSNAP$_{\text{E-37W-500m}}$.

Hence the decline in net inflow (difference inflow-outflow) of these upper limb waters after the year 2000 is the result of increasing outflow and must be balanced either by reduced transformations across the $\sigma_0 = 27.65 \, \text{kg m}^{-3}$ isopycnal or a reduction in the volume of water $\sigma_0 < 27.65 \, \text{kg m}^{-3}$ (negative volume tendency) in the Labrador Sea (Fig. 13b). On these multiyear timescales, the declining net inflow (and therefore the increasing Labrador Sea outflow) of upper limb waters is found

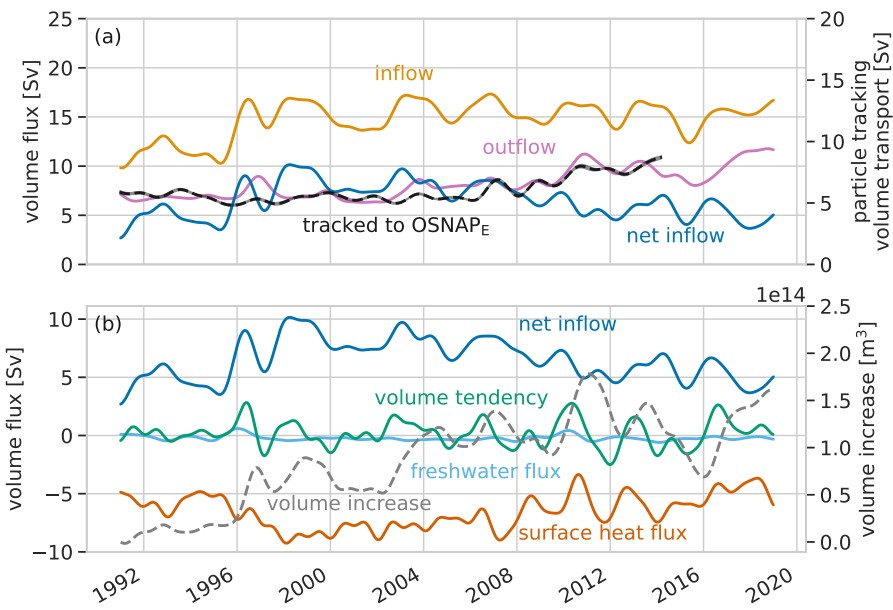

**Figure 13.** Labrador Sea water mass balance time series for $\sigma_0 < 27.65\,\mathrm{kg\,m^{-3}}$. The inflow ((**a**), light orange) shows no clear trend after 1998, whereas the outflow ((**a**), purple) shows a clear increase. Summing inflow and outflow shows a reducing net inflow of lighter waters to the Labrador Sea after 1998 (blue line, (**a,b**)). The transport leaving the Labrador Sea, heading to $\mathrm{OSNAP_{E\text{-}37W\text{-}500m}}$, from the particle tracking is included in (**a**) (dashed black line) for comparison. Note the slightly different scale on the right hand y-axis for this line to help comparison since there is not a 1:1 correspondence between water $\sigma_0 < 27.65\,\mathrm{kg\,m^{-3}}$ leaving the Labrador Sea and water tracked to $\mathrm{OSNAP_{E\text{-}37W\text{-}500m}}$). In (**b**), this reduced net inflow of light water must be balanced by water mass volume transformations – heat flux out of the ocean (orange/red line), freshwater flux (light blue) – or show as changing volume in the volume tendency term (green). Time series are generated from 5-day VIKING20X output lowpass filtered with a butterworth filter and $18\,\mathrm{month}$ cutoff.

to be due to reduced transformation to denser layers by surface heat flux (i.e. reduced cooling) along the $\sigma_0 = 27.65\,\mathrm{kg\,m^{-3}}$ isopycnal outcrop. The contributions from freshwater flux (Fig. 13b) and diapycnal mixing are smaller. The contribution from volume tendency shows variability around a mean near zero, but integrated over time leads to important longer-timescale changes in isopycnal depths in the Labrador Sea and Labrador Current (see Sections 5.3.3 and 6.3).

Prior to 2000, there is a period of increasing inflow of upper limb waters in the West Greenland Current (Fig. 13a). This increase is balanced by increasing transformation to lower limb densities within the Labrador Sea by surface heat flux, resulting in little change in outflow of upper limb waters.

In summary, in the VIKING20X model, reduced cooling in the Labrador Sea between 2000 and 2013 leads to reduced transformation to denser water and so to increased transport of lighter upper waters out of the Labrador Sea. This increased transport, when mixed with the warmer, saltier waters of Gulf Stream origin in the NAC, increases the relative contribution of fresher, cooler Labrador Sea origin waters and acts to freshen and cool the transports through the eastern subpolar North Atlantic. Note that the increasing lighter water outflow occurs at a time of relatively weak and constant (or further weakening)

SPG strength in VIKING20X (Biastoch et al., 2021). So rather than being associated with increased subpolar gyre circulation (which is mostly barotropic) the increased outflow of lighter waters is associated with reduced outflow of denser waters below.

## 6.3 Comparison with observations

In Section 5.3.3, we proposed isopycnal deepening in the outflow region as the mechanism increasing volumes of lighter waters exiting the Labrador Sea. Here (Fig 14), we compare timeseries of isopycnal depths in VIKING20X with those calculated from observations isopycnal depths in the Labrador Sea outflow region. Between the early 1990s and about 2011-12, the low-pass filtered depth of isopycnals in the range $\sigma_0 = 27.65$–$27.70\,\mathrm{kg\,m^{-3}}$ more than doubles (Fig. 14) in both model and observations. After 2013, with resumption of deeper convection in the Labrador Sea, the annual mean isopycnal depths begin to decrease.
While there is good agreement between isopycnal depth changes in model and observations on decadal timescales, at shorter interannual to pentadal timescales the agreement decreases, in particular for the lighter layers in the Labrador Sea boundary region. There are relatively fewer observations in this region, possibly leading to the higher interannual variability observed.

Using the same observational dataset, Yashayaev and Loder (2017) studied the LSW (Labrador Sea Water), LSW typically forms the intermediate water mass (approximately $\sigma_0 = 27.74$–$27.80\,\mathrm{kg\,m^{-3}}$) immediately below the upper layer waters we
are interested in here. The detailed repeat observations described show cumulative surface heat fluxes in the Labrador Sea after the early 1990s leading to increased temperatures in the depth range 200–2000 m (Yashayaev and Loder, 2017, Fig. 3), resulting in the 2012—2016 LSW class being one of the deepest and most persistent ever observed. Recent work (Volkov et al., 2022, [preprint]) also shows rising sea levels in the Labrador Sea between 1994 and 2010 associated with increased ocean heat content, due to reduced surface heat flux out of the Labrador Sea.

The preceding observational analysis, combined with the previously published observations from the region, provides convincing evidence that reduced heat loss over a period of over 10 years in the Labrador Sea led to deepening upper layers. Comparison of VIKING20X modelled Labrador Sea isopycnal depths with these observations therefore supports that VIKING20X realistically reproduces isopycnal deepening on pentadal to decadal timescales.

## 7 Conclusions

We have used the high-resolution eddy-rich nested ocean–sea-ice model VIKING20X to explore the causes of the exceptional freshening (and cooling) event observed in the surface 500–1000 m of the eastern subpolar North Atlantic in the years from 2012 onwards. We find a major cause to be reduced surface heat loss from the Labrador Sea during the decade preceding the exceptional freshening (Fig. 15).

Surface heat loss in the Labrador Sea transforms lighter inflow from the West Greenland Current to denser water which flows
out in the Labrador Current system. However, not all the light surface inflow water is converted to denser intermediate and deep water, some flows out of the Labrador Sea in the upper few hundred metres at densities $\sigma_0 < 27.65\,\mathrm{kg\,m^{-3}}$. Reducing heat loss in the Labrador Sea therefore can lead to increased volumes of lighter waters remaining in the upper layers, deepening the isopycnals and increasing the outward transport of lighter upper layer waters in the Labrador Current system.

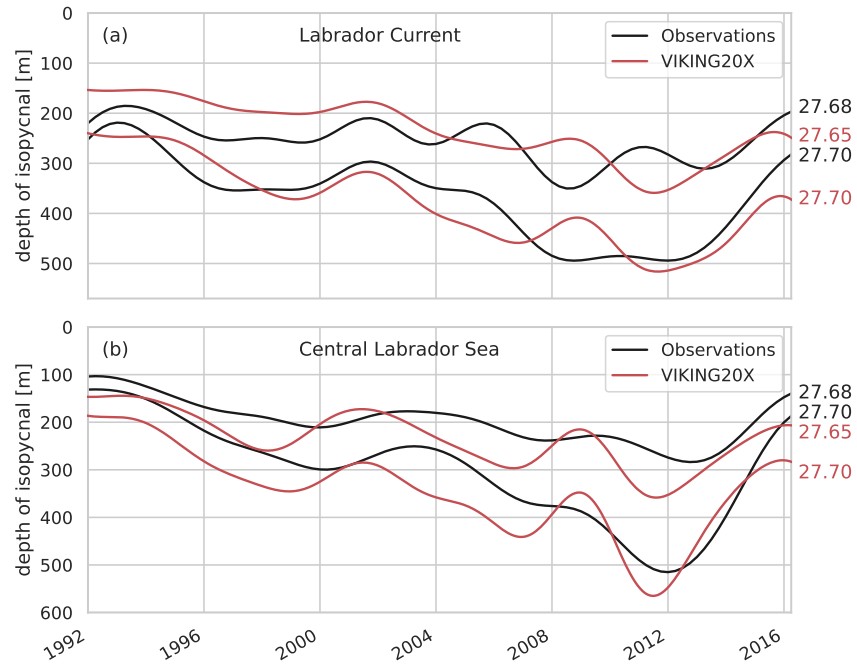

**Figure 14.** Time series of low pass filtered (2-year cutoff) mean isopycnal depth from model (red lines) and observations (black lines) in the Labrador Sea outflow region (see Fig 1). **(a)** Labrador Sea boundary region, bed depths 750–3275 m of over the Labrador Slope. **(b)** sea bed deeper than 3275 m in the central Labrador Sea. Note the slightly different isopycnals plotted. Isopycnals in the range $\sigma_0 = 27.65$–$27.70\,\mathrm{kg\,m^{-3}}$ are seen to deepen across the region between the early 2000s and the early 2010s before shallowing after 2012. The deepening of the isopycnals is associated with increased outflow of lighter upper waters from the Labrador Sea.

The relatively fresh and cold upper layer waters flowing out of the Labrador Sea combine with warmer saltier upper layer
waters from the subtropics. They then form the upper waters of the eastern subpolar North Atlantic together. The TS character-
istics of the resulting waters are largely governed by the ratio of volumes of water from the two major source regions in the mix.
In this way reduced Labrador Sea surface heat loss, and resulting outward transport increased volumes of lighter upper layer
waters from the Labrador Sea, is found to lead to freshening and cooling downstream in the eastern subpolar North Atlantic.

The quantitative particle tracking allows us to estimate contributions to the exceptional freshening and cooling. For the
salinity, we find increasing volume transport of light waters out of the Labrador Sea to account for 60 % of the freshening, with
27 % due to decreasing northward transport from the Gulf Stream (due to weakening modelled AMOC). The small remainder
is a combination of along-track external fluxes and changes in TS properties in the origin regions.

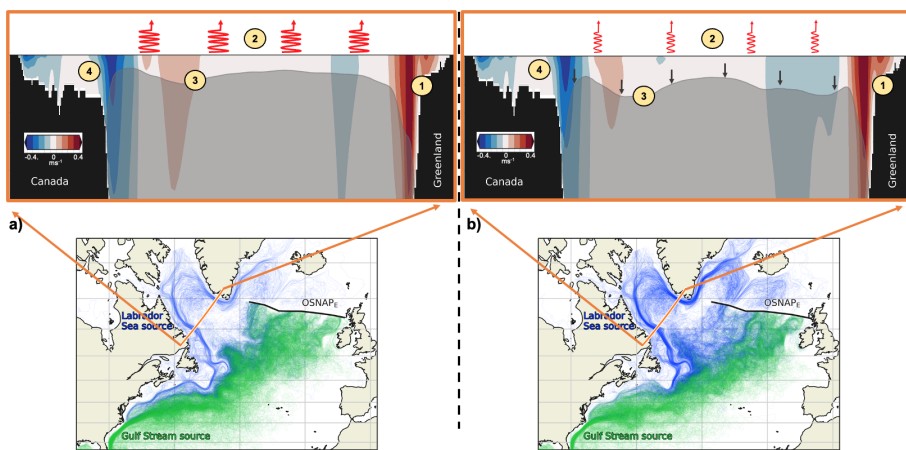

**Figure 15.** Diagram illustrating the mechanism for reduced surface heat loss leading to increased outflow from the Labrador Sea: (a) late 1990s/early 2000s for a period of higher heat loss (red zig-zag lines); (b) early 2010s for the period of reduced heat loss (small red zig-zag lines). The upper panels show water mass densities (less dense upper layer clear and denser intermediate and deep waters dark grey), and mean velocities across OSNAP$_W$ (red is northward and blue is southward). The lower panels are number of backward tracks crossing OSNAP$_W$ and show the contrasting transport pathways to OSNAP$_E$. During the two periods inflow of light waters along West Greenland (1) remains similar in thickness and velocity structure. Reduced surface heat loss (2) in the freshening period (b) leads to less transformation from light to deep water, deepening the less dense layer (3) and so increased outflow of the light waters on the Labrador Shelf (4). The lower panel shows how during the freshening event (b) the proportion of water from the Labrador Sea (blue) increases and pushes eastward at OSNAP$_E$. The Gulf Stream origin water decreases in volume and is confined more to the eastern end of the OSNAP section. While total transport from the Labrador Sea is higher in (b) than (a), transport via the loop path southward along the eastern US has decreased in (b). Most of the increased flow from the Labrador Sea is water circulating in the SPG.

For temperature, the picture is more complicated, half of the overall mean cooling of the water leaving the combined origin regions is mitigated along-track by reduced heat loss to the atmosphere from the originally cooler water. Hence, while surface salinities in the eastern subpolar North Atlantic have shown extreme freshening (the freshest in 120 years (Holliday et al., 2020)), temperatures, though cooler than average, have shown a relatively smaller decrease. The net cooling of water leaving the combined origin regions is, as for salinity, driven mostly by increasing volumes of water originating from the Labrador Sea (over 55 %) and decreasing volume of water from the Gulf Stream (20 %). In contrast to the freshening, a notable contribution to the cooling is also made by the reduced mean temperature of water from the Gulf Stream at their origin (15 %).

Bryden et al. (2020), using heat and freshwater budgets, find that reduced northwards fluxes from the subtropics to the subpolar North Atlantic can explain the bulk of the observed eastern subpolar North Atlantic cooling and freshening. In contrast, while in the current analysis these processes make a notable contribution to the exceptional freshening and cooling, we find the largest contribution to be made by variability in fluxes from the Labrador Sea.

We have considered previously published suggestions of subpolar gyre-scale mechanisms driving the exceptional freshening and cooling, specifically subpolar gyre strengthening and/or expansion (see e.g. Bersch, 2002; Sarafanov, 2009; Bersch et al.,

2007; Hátún et al., 2005; Koul et al., 2020; Kenigson and Timmermans, 2020) and the rearrangement of heat and freshwater in the upper subpolar gyre caused by changing pathways of LC-Arctic water (Holliday et al., 2020) driven by windstress curl. Our methodology, calculating mean salinity and temperature of northward transports across the whole upper eastern SPNA (rather than area average salinities and temperatures or focusing on a subregion), precludes pure subpolar gyre eastward expansion (with no new transport) as a cause of the overall cooling and freshening (though it could contribute to redistribution of heat and salt transport along the OSNAP$_E$). Examining the subpolar gyre strength (SPGI, Fig. 5), in VIKING20X-JRA-short subpolar gyre strength increases from 2014 after a period 2000–2013 with interannual variability but no significant trend. While the strengthening subpolar gyre coincides with the strongest freshening at OSNAP$_E$, time series of salinity and temperature (Fig. 7) show this freshening and cooling starting before 2012. Time series of transport variability leaving the Labrador Sea and Gulf Stream origins (Fig. 5) show that the changed relative proportions of waters in the mix are set by 2013, before the subpolar gyre strengthens. So, while subpolar gyre strengthening very likely contributes to the extreme of the freshening and cooling in 2014–2016, the processes leading to most of the freshening and cooling precede the subpolar gyre strengthening.

Our work partially supports the idea that changing pathways of water of Labrador Sea origin contribute to the freshening and cooling of the ESPNA (Holliday et al., 2020) in that we see a reduction of loop path transport in favour of an increase in direct path transport. This switch to a shorter path can produce a temporary increase in the proportion of fresher,colder Labrador Sea origin water in the mix; and the shorter, more northerly, path reduces along-track heat loss.

While the above processes probably all contribute, making this exceptional freshening event the strongest for 120 years, our analysis suggests that the largest contribution is from the increased outflow of lighter upper layer waters from the Labrador Sea due to deepening isopycnals. This outflow variability appears largely independent of both subpolar gyre strength and current pathways, relying on changing isopycnal depths rather than the surface currents measured by the sea surface height-based SPGI. Yashayaev and Loder (2017) show close correspondence between the annual density (and hence isopycnal depth) time series for upper intermediate waters in the central Labrador Sea and multi-year cumulative atmospheric forcing indices (of surface heat flux and winter NAO). This agreement provides support for a preconditioning role of cumulative atmospheric forcing (a tendency for more negative NAO states typically accompanied by reduced heat loss over the SPNA) over the decade preceding the exceptional freshening, leading to the peak in outflow of lighter upper layer waters (about 2014 in VIKING20X, Fig. 5b) and the subsequent (2016–2017) peak freshening in the eastern SPNA.

We further find the increased volumes transported from the Labrador Sea to the upper $500\,\mathrm{m}$ of OSNAP$_E$ to be primarily water recirculating in the subpolar gyre rather than Arctic waters of Hudson Bay, Davis Strait or Greenland Sea origin. These Arctic waters form the Labrador Coastal Current and the LC-Arctic outflow explored by Holliday et al. (2020), and a possible indirect route for fluxes out of the Arctic Ocean to influence the hydrography of the SPG (Kenigson and Timmermans, 2020). The increased recirculation in the subpolar gyre, in both absolute volume terms and as a proportion of the surface subpolar gyre circulation, implies longer residence times in the upper layers of the subpolar gyre. To first order, water crossing northwards across OSNAP-E in the upper layers does one of three things: crosses the GSR (where it is made denser and returns in deeper layers): is made denser within the SPG sinking to intermediate and deep layers; or recirculates in the upper SPG. To increase the proportion recirculating, one of the others needs to reduce. The most likely candidate is reduced conversion to denser

water in the SPG as exchanges across GSR appear fairly constant from observations. The implications of this for subpolar and overturning dynamics remain the subject of further work.

It is interesting to contrast these results with those of Robson et al. (2016) who also propose a link between Labrador Sea densities and cooling of the upper eastern subpolar North Atlantic. But, while Robson et al. (2016) link the eastern subpolar North Atlantic cooling to record low densities (lighter water) in the *deep* Labrador Sea (in turn reducing transport of warm water from the south associated with decreased AMOC), here we find the link is to increased volumes of lighter water in the *upper* water column.

The model results have enabled us to form a hypothesis, describing a possible mechanism linking Labrador Sea surface heat fluxes with salinity and temperature changes in the eastern Subpolar North Atlantic, initial comparison of the model output with observational data add some support for this hypothesis. VIKING20X is shown to reasonably reproduce both the strength and spatial structure of the exceptional freshening event shown in EN4 analyses (Fig. 1). Model-data comparison of vertical structure in the Labrador Sea outflow region (Fig. 14) – the region critical to our hypothesis – shows VIKING20X reproducing observed scales and timings of isopycnal deepening and shoaling in the upper 500 m.

Deepening isopycnals in the upper 500–1000 m of the Labrador Sea outflow, and the resulting increased outflow of lighter waters, has been shown here to lead to freshening and cooling of the upper eastern subpolar North Atlantic. The increased layer thickness of surface waters in the outflow must reduce the thickness of intermediate and/or deep outflows below. Both of these deeper layers also export water which is relatively fresh and cool compared to other North Atlantic water masses of similar density. While a full freshwater and heat budget of the subpolar gyre is beyond the scope of the current work, this suggests that the the exceptional freshening and cooling of the upper 500–1000 m of 2012 to 2016 may at least in part represent a change in the vertical distribution of heat and freshwater in the eastern subpolar North Atlantic. This would complement the horizontal redistribution within the subpolar gyre proposed by Holliday et al. (2020), and the south–north redistribution of Bryden et al. (2020). Some support for the vertical redistribution of heat and salt in the subpolar gyre can be seen in the vertical sections of model temperature and salinity anomalies presented here (Fig. 1e,f) and the OSNAP observational sections (Holliday et al., 2018, Fig. 6), which both show warming and salinification below the thermocline.

While the VIKING20X model used here contains a freshening and cooling signal in the upper eastern subpolar North Atlantic with many similarities to the observed signal, the correspondence is, of course, not exact. VIKING20X slightly overestimates the magnitude of the exceptional freshening event (by up to 25 %, Figs. 1,2), while also having a long-term underestimate of the AMOC strength by 10–20 % (Fig. 9). Consider an extreme scenario where all the overestimate of freshening and cooling is due to overestimate of variability of transport of Labrador Sea origin, and the contribution from variability of transports of Gulf Stream origin is too low by 10–20 %. This gives rough estimates of a minimum contribution from Labrador Sea origin transport variability of 50 % (freshening) or 45 % (cooling) and a maximum contribution from Gulf Stream origin transport of 40 % (freshening) or 25 % (cooling). We must further caveat the results presented here based on some limitations of the Lagrangian tracking methods. We do not assess possible biases either due to eddy effects and diffusion, which are not represented in the particle tracking, or due to the streamtube assumption of volume transport remaining constant along particle tracks.

*Code and data availability.* EN.4.2.2 data were obtained from https://www.metoffice.gov.uk/hadobs/en4/ and are Crown Copyright, Met Office, 2022, provided under a Non-Commercial Government Licence http://www.nationalarchives.gov.uk/doc/non-commercial-government-licence/version/2/. OSNAP data were collected and made freely available by the OSNAP (Overturning in the Subpolar North Atlantic Program) project and all the national programs that contribute to it (www.o-snap.org). Data from the RAPID AMOC monitoring project is funded by the Natural Environment Research Council and are freely available from www.rapid.ac.uk/rapidmoc (Frajka-Williams et al., 2021).

The NEMO code is available at

https://forge.ipsl.jussieu.fr/nemo/svn/NEMO/releases/release-3.6 (NEMO System Team, 2021). Our experiments are based on revision 6721. OceanParcels Lagrangian tracking code is available at https://oceanparcels.org/, our experiments used v2.2.2. The original underlying VIKING20X model output is available on request from GEOMAR research data management (datamanagement@geomar.de).

All particle tracking and analysis was done in Jupyter notebooks using Python. The main Python packages used include OceanParcels, Numpy, Scipy, xarray, matplotlib, and cartopy. All plots use seaborn styles (Waskom, 2021) with all line plots using the 'colorblind' colour palette. Filled contour plots use the cmocean oceanography colormaps (Thyng et al., 2016). The full details of all packages and version numbers can be found in the published Jupyter notebooks at https://doi.org/10.5281/zenodo.6393655. The trajectory data are available from https://hdl.handle.net/20.500.12085/830c72af-b5ca-44ac-8357-3173392f402b.

The collection and analysis of oceanographic (ship, mooring and profiling float) observations in the Labrador Sea were conducted as part of the Deep-Ocean Observation and Research Synthesis (DOORS) initiated and maintained by Igor Yashayaev (the Bedford Institute of Oceanography, Fisheries and Oceans Canada) as a follow-up to the World Ocean Circulation Experiment (WOCE). These data are available by contacting Igor Yashayaev.

*Author contributions.* ADF and SAC defined the overall research problem. ADF guided the research and methodology and performed the analyses. ADF, CS and WR designed and performed the Lagrangian experiments. SR helped to refine the analyses and interpretation of the Lagrangian experiments. NJF contributed analyses of EN4 observations in the Labrador Sea and TM contributed Eulerian analyses of VIKING20X outputs in the NAC. IY became a co-author at the Ocean Science manuscript revision stage, contributing analysis and interpretation of Labrador Sea observational data. All co-authors discussed and refined the analyses and contributed to the text.

*Competing interests.* No competing interest are present

*Acknowledgements.* ADF would like to acknowledge useful discussions with Stefan F. Gary in the early stages of this work. The ocean model simulation was performed at the North German Supercomputing Alliance (HLRN) and on the Earth System Modelling Project (ESM) partition of the supercomputer JUWELS at the Jülich Supercomputing Centre (JSC). We thank the NEMO system team for support. The trajectory simulations were conducted at the Christian-Albrechts-Universität zu Kiel (NESH).

*Financial support.* We acknowledge funding from the U.K. Natural Environment Research Council (NERC) National Capability programs CLASS (NE/R015953/1), ACSIS (NE/N018044/1), and NERC grants U.K. OSNAP Decade (NE/T00858X/1, NE/T00858X/2, NE/T008938/1) and SNAP-DRAGON (NE/T013400/1, NE/T013494/1). Additional support was received from the European Union Horizon 2020 research and innovation program under grants 727852 (Blue-Action), 818123 (iAtlantic). This output reflects only the author's view and the European Union cannot be held responsible for any use that may be made of the information contained therein. We also acknowledge support from the German Federal Ministry of Education and Research (grant no. SPACES-CASISAC (03F0796A)).

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
