# Peer review of "Exceptional freshening and cooling in the eastern subpolar North Atlantic caused by reduced Labrador Sea surface heat loss"

_Ocean Science, 2022_

## Referee Comment (RC1)

The study uses the VIKING20X-JRA-short ocean hindcast simulation to explore mechanisms explaining the anomalously fresh and cold conditions in the subpolar North Atlantic during 2012-2016. Utilizing Lagrangian tracking, the authors show that the freshening/cooling seen at OSNAP-East to a large extent is caused by a higher fraction of Labrador Sea source water due to increased recirculation in the subpolar gyre. The study addresses a timely question, and makes for an interesting contribution in the debate regarding subpolar variability. That being said, I have several questions and comments related to the mechanism proposed and how it relates to previous studies. Moreover, the manuscript is quite dense with results, and being precise with definitions and language will be crucial for getting the main messages across to a reader.

Mechanisms and relationship to related work

1)  The introduction highlights particularly the 'warming hole'. However, as the warming hole is a multi-decadal signal thought to be caused by effects that are not the main focus here (Keli et al. 2020; cloud feedback, low-latitude AMOC), the introduction reads a little misleading. I recommend restructuring so that the 2012-2016 freshening and cooling is the focus, as well as previously proposed mechanisms for interannual to decadal variability (i.e. expand on mechanisms mentioned in l.34-42 and in Section 5).

2)  Throughout, it's unclear to me exactly how the proposed mechanism agree/disagree/link to previous proposed mechanisms in literature. Formulations such as l.575 'we find neither of these to be fully convincing' make it sound like you discard all previous work, but surely your mechanism is not entirely independent of what's been proposed earlier?

3)  l.314-316, l.322-324 and Fig. 8: You do see signs of an eastward shifted subpolar front, but you seem quite dismissive of horizontal redistribution in the conclusion (l.600-601). In addition to Holliday et al. 2020 and Desbruyères et al. 2021, Kenigson et al. 2020 and Asbjørnsen et al. 2021 are relevant for such horizontal shifts. Asbjørnsen et al. 2021 is also quite similar in terms of the particle tracking approach.

4)  Are you seeing the subpolar trend reversal in VIKING20X as documented in Desbruyères et al. 2021?

5)  Fig. 8a-b: Eastern and western parts look quite different. Interesting that OSNAP-E-37W-500m total (Fig. 5a) is dominated by what's happening at 37W-21W. A point that should be noted? The fractional evolution seen for the eastern part is more consistent with what is seen for the Nordic Seas inflow in Asbjørnsen et al. 2021.

6)  l.358-360 and onwards: In any sort of AMOC change discussion, it needs to be clear that there is no consensus on whether the AMOC actually has systematically weakened over the 20th century until today (e.g., debate summarized in Jackson et al. 2022, Latif et al. 2022).

7) l.445-446: Labrador Coastal Current and 'the other' – are both these branches what's typically called the Labrador Current? I also don't get the LC-Arctic and LC-Atlantic distinction (l.411-412).

8) You make a convincing case for the freshening/cooling seen at OSNAP-E is due to a higher fraction of Lab. Sea water (Fig. 5, Fig. 7, Table 1) related to longer residence times in the SPG (Fig. 10a). I don't fully get why more SPG recirculation necessarily must be due to reduced heat loss and deepening isopycnals (l.435-441). Do you reference studies showing this mechanism for the SPG anywhere? Perhaps this point needs to be repeated in the conclusions.

Lagrangian analysis:

1) Are particles released over the full OSNAP-East line as said in l.75-76 or only over the part of OSNAP-East shown in Figure 3? Are you analysing releases over the upper 1000m as said in l. 76 or the upper 500m as stated in l.111. I think the information in l.111-116 needs to come earlier to avoid conflicting messages.

2) l.84-86: A bit unclear. How about explaining it as: 'Each particle represents a volume transport of 0.001797 Sv. The number of particles released along the 2-dimensional OSNAP-E section is scaled with the model velocity normal to the section at the release time.'

3) l.87-89: What is the reasoning for choosing Parcels and not a tool like Ariane where streamlines are computed analytically?

4) l.89-95: I don't quite get the type of errors discussed here (Errors induced by temporal resolution? Errors related to lacking diffusion? Errors related to particle release number? Errors related to assumption of stationarity?). What are the 32 subsets?

5) Gulf Stream and Lab. Sea are defined as the two 'upstream origins' for OSNAP-E water (l.96). Then you subdivide Lab. Sea 'origin' by 'upstream origin' again (l.102): Hudson Bay, Davis Strait. Greenland Sea, Denmark Strait, SPG. Later in the manuscript the word 'source' is frequently used. I would be very clear with the definitions and use them consistently throughout. Perhaps Lab. Sea and Gulf Stream is your 'upstream sources' while the Labrador source region is subdivided by four 'origin regions'?

6) Figure 3: I'm confused by the lines on the map. Lines over land should be gone? Are particles crossing the orange line cutting across the Lab. Sea defined as having Lab. Sea origin? Is the north-south orange line the Loop path/Slope Sea pathway definition? I get the east-west green line is the Gulf Stream definition, but what about the north-south thin green line? What about showing definitions for Davis, Hudson, Greenland Sea origins in panels f-h, respectively?

7) l.209-212: Such fractions will depend on the depths evaluated over and the source definition used. In Koul et al. 2020, particles are initialised in the upper 100m only. In Asbjørnsen et al. 2021, 26% of the Iceland-Scotland Ridge inflow (upper 1000m or so) has a subpolar or Arctic origin (Davis, Hudson, Denmark straits, or circulation in the SPG), with 42% of the surface inflow being water from Davis Strait and Hudson Bay.

Minor comments / technical corrections:

1) l. 80: reference in parenthesis or write 'shown in Biastoch et al. 2021'.

2) Fig. 1 and Fig. 2 is not referenced in the text until page 9. Either reference earlier or rethink figure-order.

3) l.107-110: Would shorten this point. Doesn't need to be its own paragraph.

4) Section 2.4: Unless you want to expand on OSNAP measurements and how the EN4 product is produced, I would delete this section and move the sentence somewhere suitable.

5) Section 2.5: Not really necessary information – I recommend deleting the section and put the information about code availability in the data statement at the end.

6) l.216-217: Stated already in section 2.2, not necessary to repeat.

7) Starting sentences with 'So' or 'But' (sometimes) making them not 'full sentences' with a subject, verb, and an object: e.g., l.220, l.247, l.255, l.269, l.427, l.465, l.519, l.546.

8) l.220-223: No need to repeat the details here.

9) l.294-305: I would cut this section and save such summarizing statements for the conclusion. If you are worried some of the interesting results might be 'lost' because the manuscript is quite dense, you could even do a bullet point list summarizing the main results in the conclusion.

10) l.337-340: check punctuation and parentheses.

11) l.341-344, l.411:413: check parentheses.

12) Section 5.3 title: title sounds like you again will discuss what was addressed in 5.1. I would find an alternative pointing to the Labrador Sea focus.

13) l.414-416: ref. Fig. 10?

14) Fig. 10 and l.414-419: What depth is the 'light upper layers' - are you still looking at 0-500m?

15) l.528-532: Too long sentence – difficult to decipher.

16) Legends Fig. 7, Fig. 8, Fig. 10 looks quite messy. I would display text as a structured column. Fig. 8 could have one common legend on the side.

Mentioned literature:

Kenigson et al. 2020 – 10.1175/JPO-D-20-0071.1
Asbjørnsen et al. 2021 – 10.1175/JCLI-D-20-0917.1
Desbruyères et al. 2021 - 10.1038/s43247-021-00120-y
Jackson et al. 2022 - 10.1038/s43017-022-00263-2
Latif et al. 2022 - 10.1038/s41558-022-01342-4

---

## Author Comment (AC1)

**OS-2022-18: Author Comment to Anonymous Referee #1**

We thank Referee #1 for their positive comments and for recognising the current work as an interesting and timely contribution. We also thank Referee #1 for their constructive criticism and here we respond to their points and outline the changes we will make to address the comments if invited to submit a revised manuscript.

**Referee's comments are in plain text, authors' comments in response are in **bold**.**

**Anonymous Referee #1:**

The study uses the VIKING20X-JRA-short ocean hindcast simulation to explore mechanisms explaining the anomalously fresh and cold conditions in the subpolar North Atlantic during 2012-2016. Utilizing Lagrangian tracking, the authors show that the freshening/cooling seen at OSNAP-East to a large extent is caused by a higher fraction of Labrador Sea source water due to increased recirculation in the subpolar gyre. The study addresses a timely question, and makes for an interesting contribution in the debate regarding subpolar variability. That being said, I have several questions and comments related to the mechanism proposed and how it relates to previous studies. Moreover, the manuscript is quite dense with results, and being precise with definitions and language will be crucial for getting the main messages across to a reader.

Mechanisms and relationship to related work

The introduction highlights particularly the 'warming hole'. However, as the warming hole is a multidecadal signal thought to be caused by effects that are not the main focus here (Keli et al. 2020; cloud feedback, low-latitude AMOC), the introduction reads a little misleading. I recommend restructuring so that the 2012-2016 freshening and cooling is the focus, as well as previously proposed mechanisms for interannual to decadal variability (i.e. expand on mechanisms mentioned in I.34-42 and in Section 5).

**Both Referees#1 and #3 felt that the introduction wrongly emphasises the multi-decadal 'warming hole' over the exceptional freshening and interannual to decadal variability. We will restructure the introduction to address this.**

Throughout, it's unclear to me exactly how the proposed mechanism agree/disagree/link to previous proposed mechanisms in literature. Formulations such as I.575 'we find neither of these to be fully convincing' make it sound like you discard all previous work, but surely your mechanism is not entirely independent of what's been proposed earlier?

We will include more discussion on how our proposed mechanism relates to previous proposed mechanisms, at the relevant points particularly in sections 5 and 6. Although it was not our intention to discard other work, we agree that we need to be more careful with our language and will modify this, while recognising that the reason for embarking on this study was the idea that the previously published work did not offer a complete explanation of the freshening event.

I.314-316, I.322-324 and Fig. 8: You do see signs of an eastward shifted subpolar front, but you seem quite dismissive of horizontal redistribution in the conclusion (I.600-601). In addition to Holliday et al. 2020 and Desbruyres et al. 2021, Kenigson et al. 2020 and Asbjørnsen et al. 2021 are relevant for such horizontal shifts. Asbjørnsen et al. 2021 is also quite similar in terms of the particle tracking approach.

We think this is a fair point and we will revise the conclusion to endeavour not to be dismissive of horizontal redistribution. This passage was intended to be quite speculative, postulating that while horizontal redistribution is clearly occurring (see figure 1) it probably isn't the whole story and the mechanism identified here provides a potential route to additional vertical redistribution of heat and salt.

An interesting additional perspective on both this question and the one above is provided in a recent manuscript of Volkov et al. (Ocean Science Discussions, *preprint*,

https://egusphere.copernicus.org/preprints/2022/egusphere-2022-354/). They show both reduced heat loss associated with anomalous positive SSH anomalies in the Labrador Sea (associated weaker western SPG and deeper isopycnals), and that widespread cooling and freshening in the eastern SPG must be predominantly advective. Here we link those two processes.

Are you seeing the subpolar trend reversal in VIKING20X as documented in Desbruyres et al. 2021?

We possibly begin to see some reversal of the subpolar trend (see e.g. Figure 7). It looks like this may be later than in Desbruyres et al., but our section is at the northern edge of the area they consider.

Fig. 8a-b: Eastern and western parts look quite different. Interesting that OSNAP-E-37W-500m total (Fig. 5a) is dominated by what's happening at 37W-21W. A point that should be noted? The fractional evolution seen for the eastern part is more consistent with what is seen for the Nordic Seas inflow in Asbjørnsen et al. 2021.

We will note this in the text. The dominance of western segment over the eastern segment is due to the larger volumes transported through the western segment, and may also be change dominating visually over no change. Asbjørnsen et al. (2021) consider the flow across the GSR to the Nordic Sea, the fractional evolution for this is probably more consistent with the eastern part because transport through GSR is dominated by Rockall Trough/Rockall-Hatton Bank transports. Iceland Basin and Irminger Sea northward transports, while larger, are more likely to recirculate and be transformed to deeper waters south of the GSR.

I.358-360 and onwards: In any sort of AMOC change discussion, it needs to be clear that there is no consensus on whether the AMOC actually has systematically weakened over the 20th century until today (e.g., debate summarized in Jackson et al. 2022, Latif et al. 2022).

**Agreed. We will mention this in section 5.2 but we should also point out that long-term AMOC changes are not actually relevant to the sub-decadal scale changes we consider.**

I.445-446: Labrador Coastal Current and 'the other' – are both these branches what's typically called the Labrador Current? I also don't get the LC-Arctic and LC-Atlantic distinction (I.411-412).

I.445-446: Yes.

I.411-412 The -Arctic and -Atlantic suffices in LC-Arctic and LC-Atlantic refer to the immediate upstream source of water in the Labrador Current. LC-Arctic has come south through the Davis Strait. LC-Atlantic water has predominantly come into the Labrador Sea from the Irminger Basin in the West Greenland Current. On reflection we concentrate on our source-based categories and introduce the LC-Arctic and LC-Atlantic terminology only to orient the current work in the context of that which has gone before.

You make a convincing case for the freshening/cooling seen at OSNAP-E is due to a higher fraction of Lab. Sea water (Fig. 5, Fig. 7, Table 1) related to longer residence times in the SPG (Fig. 10a). I don't fully get why more SPG recirculation necessarily must be due to reduced heat loss and deepening isopycnals (I.435-441). Do you reference studies showing this mechanism for the SPG anywhere? Perhaps this point needs to be repeated in the conclusions.

L.435-441: This paragraph is speculative, and partly a trailer for what is to come in section 6. It is just expressing a volume transport budget. The idea is that upper water crossing northwards across OSNAP-E does one of three things: crosses the GSR (where it is made denser and returns in deeper layers): is made denser within the SPG sinking to intermediate and deep layers; or recirculates in the SPG. To increase the proportion recirculating, one of the others needs to reduce. The most likely candidate is reduced conversion to denser water in the SPG as flows across GSR appear pretty constant from observations. We will clarify this here with appropriate references. We agree it also needs repeating in the conclusions.

**Lagrangian analysis:**

Are particles released over the full OSNAP-East line as said in I.75-76 or only over the part of OSNAP-East shown in Figure 3? Are you analysing releases over the upper 1000m as said in I. 76 or the upper 500m as stated in I.111. I think the information in I.111-116 needs to come earlier to avoid conflicting messages.

Agreed. We will clear this up. We did release more particles than we used in the analysis. The reader does not need to know this though. Particularly those particles west of 37W. We will present some headline results from the particles down to 1000 in response to comments by other referees, so will keep the information about the particles released down to 1000 m.

I.84-86: A bit unclear. How about explaining it as: 'Each particle represents a volume transport of 0.001797 Sv. The number of particles released along the 2-dimensional OSNAP-E section is scaled with the model velocity normal to the section at the release time.'

**Agreed.**

I.87-89: What is the reasoning for choosing Parcels and not a tool like Ariane where streamlines are computed analytically?

We don't believe the choice of tool affects the results and Parcels was chosen for pragmatic and operational reasons. Published experiments with Ariane and Parcels show very similar behaviour with appropriate choice of parameters (timesteps). E.g. Schmidt et al 2021.

I.89-95: I don't quite get the type of errors discussed here (Errors induced by temporal resolution? Errors related to lacking diffusion? Errors related to particle release number? Errors related to assumption of stationarity?). What are the 32 subsets?

We attempt to clarify this description further in the text. The errors we quantify are those due to using finite particle numbers. As we increase particle numbers each metric we discuss will converge towards a 'correct' value. Using random release positions allows us to split the particle set into a number of subsets (32 here), each of which is still random but with each particle representing a (32 times) larger transport. The spread of the estimates from these subsets gives us an idea that we have released enough particles for these sampling errors to be small compared to the signal we are examining. We do not attempt to quantify errors due to lacking diffusion in the tracking, model biases, assumption of stationarity. We include mention of these error sources in a wider 'disclaimer' requested by Referee #1.

Gulf Stream and Lab. Sea are defined as the two 'upstream origins' for OSNAP-E water (I.96). Then you subdivide Lab. Sea 'origin' by 'upstream origin' again (I.102): Hudson Bay, Davis Strait. Greenland Sea, Denmark Strait, SPG. Later in the manuscript the word 'source' is frequently used. I would be very clear with the definitions and use them consistently throughout. Perhaps Lab. Sea and Gulf Stream is your 'upstream sources' while the Labrador source region is subdivided by four 'origin regions'?

**We will make these definitions and uses consistent.**

Figure 3: I'm confused by the lines on the map. Lines over land should be gone? Are particles crossing the orange line cutting across the Lab. Sea defined as having Lab. Sea origin? Is the north-south orange line the Loop path/Slope Sea pathway definition? I get the east-west green line is the Gulf Stream definition, but what about the north-south thin green line? What about showing definitions for Davis, Hudson, Greenland Sea origins in panels f-h, respectively?

**We have hopefully cleared this up with the production of a revised figure 3. We include a draft as Figure 3.1 below.**

I.209-212: Such fractions will depend on the depths evaluated over and the source definition used. In Koul et al. 2020, particles are initialised in the upper 100m only. In Asbjørnsen et al. 2021, 26% of the Iceland-Scotland Ridge inflow (upper 1000m or so) has a subpolar or Arctic origin (Davis, Hudson, Denmark straits, or circulation in the SPG), with 42% of the surface inflow being water from Davis Strait and Hudson Bay.

**We include fractions calculated over different depth ranges: top 500m, top 1000m, amoc upper limb (as described in section 2.2) and expand the discussion to include Asbjørnsen et al. 2021 numbers.**

Minor comments / technical corrections:

1. I.80: reference in parenthesis or write 'shown in Biastoch et al. 2021'.

ОК

2. Fig. 1 and Fig. 2 is not referenced in the text until page 9. Either reference earlier or rethink figure-order.

**These figures are mentioned first in the introduction. See L33.**

3. I.107-110: Would shorten this point. Doesn't need to be its own paragraph.

4. Section 2.4: Unless you want to expand on OSNAP measurements and how the EN4 product is produced, I would delete this section and move the sentence somewhere suitable.

**This section will be longer with the inclusion of details of the Labrador Sea data used in the revised model-data comparison in section 6.**

5. Section 2.5: Not really necessary information – I recommend deleting the section and put the information about code availability in the data statement at the end.

**I (AF) think this is important information for openness and reproducibility. We move it to the code availability statement.**

6. I.216-217: Stated already in section 2.2, not necessary to repeat.

**Deleted**

7. Starting sentences with 'So' or 'But' (sometimes) making them not 'full sentences' with a subject, verb, and an object: e.g., I.220, I.247, I.255, I.269, I.427, I.465, I.519, I.546.

**We've checked for and removed these minor, but distracting, style and English usage issues**

8. l.220-223: No need to repeat the details here.

**Deleted**

 I.294-305: I would cut this section and save such summarizing statements for the conclusion. If you are worried some of the interesting results might be 'lost' because the manuscript is quite dense, you could even do a bullet point list summarizing the main results in the conclusion.

We think this section is useful for readability, summarizing results from quite a dense section before moving on to new sections of results/discussion which draw heavily on section 4. We have decided to leave it in place.

- 10. I.337-340: check punctuation and parentheses.
- 11. I.341-344, I.411:413: check parentheses.

**ОК**

12. Section 5.3 title: title sounds like you again will discuss what was addressed in 5.1. I would find an alternative pointing to the Labrador Sea focus.

**ОК**

13. I.414-416: ref. Fig. 10?

**ОК**

14. Fig. 10 and I.414-419: What depth is the 'light upper layers' - are you still looking at 0-500m?

These are still the particles we have tracked backwards from the top 0-500m at OSNAP. Further upstream they are not strictly confined to the surface 500m. As the subpolar region is

ОК

predominantly an area of sinking, particles launched in the upper layers and tracked backwards will generally stay in the upper layers. We will make this clearer in the text.

15. I.528-532: Too long sentence – difficult to decipher.

Section 6.3 will be rewritten with different observational data. See the response to Referee #1, general comment 3 for more details. Draft new Figure 14 is attached below (as Fig 14.1). So this comment has been superseded.

16. Legends Fig. 7, Fig. 8, Fig. 10 looks quite messy. I would display text as a structured column. Fig. 8 could have one common legend on the side.

No change. I (AF) disagree. I find it difficult switching between lines and legends to repeatedly remind myself which line is which. Labels by the lines on the plots are much easier.

Mentioned literature:

Kenigson et al. 2020 – 10.1175/JPO-D-20-0071.1

Asbjørnsen et al. 2021 – 10.1175/JCLI-D-20-0917.1

Desbruyres et al. 2021 - 10.1038/s43247-021-00120-y

Jackson et al. 2022 - 10.1038/s43017-022-00263-2

Latif et al. 2022 - 10.1038/s41558-022-01342-4

**References:**

Schmidt, C., F. U. Schwarzkopf, S. Rühs, and A. Biastoch, 2021: Characteristics and robustness of Agulhas leakage estimates: an inter-comparison study of Lagrangian methods, *Ocean Sci.*, 17, 1067–1080, https://doi.org/10.5194/os-17-1067-2021.

---

## Author Comment (AC2)

**OS-2022-18: Author Comment to Anonymous Referee #2**

Referee's comments are in plain text, authors' comments in response are in **bold**.

First of all we thank Referee #2 for their constructive comments.

**Anonymous Referee #2**

This article sheds new light on the main drivers behind the cooling and freshening recently experienced in the upper eastern subpolar North Atlantic. To this end, the authors analyze the outputs of a historical hindcast (years 1980 to 2019) performed with the eddy-rich ocean–sea-ice model VIKING20X, using a variety of techniques and diagnostics, from lagrangian particle tracking, to water mass transformation analyses.

The article is well-timed and well written, the methodology applied is sound, figures are clear, and many of the results will be of high interest to the climate community at large, and to anyone with specific interests in the recent changes experienced in the North Atlantic and its surroundings.

I recommend a minor revision of the manuscript and enclose a list of comments that the authors would need to address to render the article suitable for publication in *Ocean Science*.

**General Comments:**

 Agreement of the VIKING20X-JRA hindcast with observations is mentioned several times throughout the text, but it is mostly derived from visual comparisons, which can be misleading. In some cases the agreement is clear, but in other cases is much less evident (see comments #4, #16 and #18 further down). Supporting these statements with some specific metrics, like linear correlations between the hindcast and the observations, will help to determine more precisely to what extent they agree with each other.

Model verification has been done extensively in Biastoch et al. (2021) which we refer to frequently throughout the manuscript. We do not want to repeat that verification here. The purpose of the initial comparison between the model and EN4 analysis (figures 1 and 2) is to show that the VIKING20X model produces a freshening and cooling signal analogous to, if not exactly the same as, that in observations, to support the use of the model to investigate this phenomenon. We feel that the visual comparisons presented are sufficient for this purpose and don't believe that further quantitative metrics comparing hindcast and observations are necessary. We will try to make the limitations of the use of an imperfect model, and the fact that what we present is a model-based hypothesis rather than objective truth, clearer in the discussion, as requested in a similar point made by Referee #3.

While the EN4 analysis is of sufficient quality and resolution for the initial comparison aimed at supporting the use of VIKING20X in this study, we find it to be of insufficient quality for the more detailed comparison required in the Labrador Sea in section 6.3, resulting in the, fully justified, criticisms of this comparison by all Referees. Here, we completely rewrite the comparison, using observational data (rather than the EN4 analyses) in the Labrador Sea. The new dataset was provided by Igor Yashayaev, tailored specifically to our Labrador Current outflow region and based on the latest version of the data presented here: https://waves-vagues.dfo-

mpo.gc.ca/Library/40974698.pdf. The comparison between model and observations here is much closer using this dataset than with EN4, with the observational data clearly showing deepening isopycnals. We believe the difference between this observational data and EN4 analysis to be largely due to the higher resolution and more focussed area used, and to sparse salinity data and relaxation to climatology in EN4 in the earlier part of the period studied making the use of the EN4 analysis here inappropriate. While these remain purely visual comparisons, we hope that because the similarities are now clearer this will allay the Referee's concerns.

• Given the comprehensive list of processes and mechanisms that are analyzed in the paper, which in many cases are interconnected, it would be very useful to include at the end of the article a schematic figure summarizing the chain of events that give rise to the freshening and cooling of the eastern subpolar North Atlantic, as supported by the model analysis.

**This is a nice suggestion and we will add appropriate schematics/diagram. (Figure 15, a draft attached below).**

• Several figures from other articles are cited throughout the text, urging the readers to keep jumping from one article to another, and thus hindering the overall readability of the article. Some of those figures include indices that could be easily incorporated in others figures of this manuscript (see comments #6, #10 and #11), making intercomparison between those and the VIKING20X indices much more straightforward. I strongly recommend the authors to include them.

**Agreed. We will include the indices from other articles in our plots where possible**

**Specific Comments:**

 [Lines 18-19] This sentence seems incomplete and inaccurate. It should (1) mention that there is a cooling on the eastern subpolar North Atlantic, not a surface warming anomaly, and (2) it should also specify that this is simulated in response to a "weakening" of the ocean circulation. Also, I recommend the authors to avoid the use of the term "predict" in the sentence, as "predictions" generally refer to historical simulations initialized from observations to phase the model with the observed internal variability. However, the warming hole is a feature that consistently appears in uninitialized historical simulations, and is therefore deemed to be mostly externally forced.

In common with suggestions from the other reviewers we have focussed the introduction on the short-term exceptional freshening and cooling event, removing the unnecessary and confusing references to the multi-decadal warming hole

2. [Line 108] It is unclear which "common technique" it refers to.

**Agreed, this will be clarified.**

3. [Figure 4b, caption] Is it "number of days" or "number of years"?

**Years. We will correct the caption.**

4. [Lines 177-181, Figure 2] While the simulated upper temperature variability in the hindcast shows a good agreement with the observed timeseries from both OSNAP and EN4 products, this is not the case for the simulated salinity. This is particularly clear for EN4, which can be compared for a substantially longer period, showing large discrepancies in terms of both the high and the low-frequency variability. Indeed, a close inspection to Figure 2 reveals that the

differences between EN4 and VIKING20X-JRA are not stationary in time, and reflect more than the systematic mean state bias stated in the text. It is true that salinity observations are quite scarce and therefore objective analyses like EN4 are subject to large uncertainties, but it remains to be checked whether the simulated variability is within the range of the observed uncertainty.

We have replaced figures 1 and 2 with a revised comparison (figures 1.1 and 2.1 below). We now compare x-z area averages of temperature and salinity rather than depth-averages and include the error bars on the EN4 data. We also start the comparison at year 2000, as prior to the early 2000s salinity data were very sparse and EN4 analysis largely reflects the relaxation to climatology. Additionally, EN4 is a lot smoother both in space (see Fig. 1) and time. Even when averaging in space (Fig. 2) or time (Fig 1) there will always be mesoscale structure in VIKING20X output that shows as variability exceeding that of EN4. As stated in response to the general comments above, we believe that these comparisons demonstrate that the model shows sufficient skill in reproducing the particular freshening event to justify its use here. We add consideration of the implication of model skill on the conclusions we have about the mechanisms involved in the discussion.

5. [Lines 311-313] What do you mean by "contrasting evolution of transit times"? Are their associated histograms (like in Figure 4) substantially different? And in which way?

This point was not about the histograms in Figure 4, but about the time series in Figure 8. Particles crossing OSNAP-E in the west showed marked lengthening of transit times (and longer track lengths and slower mean speeds) in the 2000s, contrasting with particles crossing further east which showed much steadier speeds and transit times. This will be clarified in the text.

6. [Lines 334-340] Instead of referring the reader to Figure 10 in Biastoch et al 2021, it would be preferable if the authors included the SPG index in a subpanel of Figure 9, where it can be directly compared with the AMOC indices.

**This is a good idea and will be done.**

- 7. [Lines 339-340] The two parentheses referring to Figures 5 and 7 need to be closed.
- 8. [Lines 363-364, Figure 9] Why did you choose to plot the AMOC at 29°N and not at the same latitude of the RAPID array (i.e. 26°N)? It would be indeed very interesting to include a direct comparison of the hindcast with the RAPID data, to learn more about the model realism in representing dynamical aspects, like the North Atlantic ocean circulation.

This direct comparison between VIKING20X AMOC and the RAPID array is in Biastoch et al. (2021), referred to in the text, and does not need to be repeated here. On the choice of latitude 29°N for the calculations, we will add brief explanation to the text, and include more details here. There is very little, if any, difference between AMOC at 26°N and 29°N and the choice was made entirely for ease of calculation.

We chose 33°N for the Lagrangian analysis because it was the furthest north it could be (so keeping tracking times as short as possible) and be confident that particles caught crossing there were from the western STG rather than recirculating in the inter-gyre region for example.

However, for the Eulerian analysis we chose 29°N because the flow there is really simple. It lies at about the central latitude of the STG. As it is clear of the Antilles (unlike 26°N), practically all the northward upper layer flow sits on the wide 1000m deep shelf. Further offshore, almost all the

DWBC is above the slope at the edge of this shelf, clearly separate in the horizontal from the northward WBC. Then further east still is the broad southward flow in the eastern STG. All three of these are clearly separate in the horizontal, making calculation of STG strength, AMOC, DWBC transport and STG to SPG surface flow all straightforward. Matching the Eulerian section with the Lagrangian at 33°N would make these calculations more complex, with some northward WBC currents overlying the southward DWBC, and many more eddies and recirculations east of the Gulf Stream. Matching with Rapid at 26°N was attractive, but the comparison is not our focus, is included in Biastoch et al. (2021), and again the flow is more complex at 26°N than at 29°N, with the WBC divided between the Florida Straight and the Antilles current, with the latter overlying the DWBC.

9. [Lines 366-368] I would not say that the AMOC is responsible for a reduction in Gulfstream transport. The Gulfstream does not respond to the AMOC, it is a component of the AMOC and as such contributes to its variability, not the other way around.

**Variability in the GS can be 'associated with' a combination of AMOC and STG variability, as can be seen in Fig 9. The text specifies the GS is *associated* with reduced flows (aka the AMOC), which is not a causal statement. We will seek to clarify this distinction in the text.**

- 10. [Lines 371-372] This can be directly shown to the reader if, as suggested in Comment #6, the SPG index is included in Figure 9.
- 11. [Lines 378-380] Here you compare again with an observed index from another article that could be easily included in this one (in Figure 9).

**We will include these indices in our plots.**

12. [Line 386] I do not find this summary statement fully justified. There has been no specific analysis in section 5.2 to rule out the AMOC weakening as a main contributor to the freshening of the eastern subpolar North Atlantic.

Agreed, we will clarify this. The analysis showing that reduced transport from the Gulf Stream source was not the majority contributor to the freshening was in section 4.3, but the contribution was still estimated at a sizeable minority of 30%. Section 5.2 confirms that this reduced contribution from the Gulf stream source is reduced AMOC.

13. [Line 394] The parenthesis needs to be closed.

**Done**

- 14. [Lines 520-521] It is important to specify that the doubling of the annual mean depth of those isopycnals only occurs in VIKING20X-JRA. In EN4, the increase in the depth of the isopycnals is very subtle and only discernible for  $\sigma o > 27.65$  kg/m3.
- 15. [Lines 526-528] Figure 14 a-c  $\rightarrow$  Figure 14c (as this is the only panel showing the isopycnals)
- 16. [Lines 537-539] From Figure 14 it is not possible to say if the model realistically represents the variability in salinity and density. Indeed, as mentioned in Comment #14, the strong deepening of the isopycnals in VIKING20X-JRA (one of the major reasons behind the freshening in the eastern subpolar North Atlantic identified in the paper) is not seen in EN4, and in particular in the upper layers.

We will completely rewrite this section based on comparison with a more appropriate Labrador Sea dataset and a new Figure 14 (preliminary version attached below as Figure 14.1). For more details see the responses to Referee #1.

17. [Lines 544-545] Can you explain why it would be counter-intuitive?

This was simply that we suggest *warming* in the surface Labrador Sea leads, purely by internal ocean advection, to *cooling* in the eastern subpolar gyre. But we rephrase to reduce confusion.

18. [Lines 589-593] Not all the model results are fully supported by observational data, in particular the deepening of the isopycnals in the Labrador Sea. Furthermore, the evolution of the Labrador Sea vertical structure only compares well with the EN4 data (Figure 14) for temperature. The agreement with the observed salinity and density is much more limited. It would be worth discussing whether, and if yes, in which sense, this poor agreement affects the main findings derived from the VIKING20X-JRA analysis.

See the response to points #14-16 above.

New Figures:

---

## Author Response (AR1)

**OS-2022-18: Author responses to Referees**

**Referee's comments are in plain text, authors' comments in response are indented in bold.**

**Anonymous Referee #1:**

> **We thank Referee #1 for their positive comments and for recognising the current work as an interesting and timely contribution. We also thank Referee #1 for their constructive criticism and here we respond to their points and outline the changes we will make to address the comments if invited to submit a revised manuscript.**

The study uses the VIKING20X-JRA-short ocean hindcast simulation to explore mechanisms explaining the anomalously fresh and cold conditions in the subpolar North Atlantic during 2012-2016. Utilizing Lagrangian tracking, the authors show that the freshening/cooling seen at OSNAP-East to a large extent is caused by a higher fraction of Labrador Sea source water due to increased recirculation in the subpolar gyre. The study addresses a timely question, and makes for an interesting contribution in the debate regarding subpolar variability. That being said, I have several questions and comments related to the mechanism proposed and how it relates to previous studies. Moreover, the manuscript is quite dense with results, and being precise with definitions and language will be crucial for getting the main messages across to a reader.

Mechanisms and relationship to related work

The introduction highlights particularly the 'warming hole'. However, as the warming hole is a multi-decadal signal thought to be caused by effects that are not the main focus here (Keli et al. 2020; cloud feedback, low-latitude AMOC), the introduction reads a little misleading. I recommend restructuring so that the 2012-2016 freshening and cooling is the focus, as well as previously proposed mechanisms for interannual to decadal variability (i.e. expand on mechanisms mentioned in l.34-42 and in Section 5).

> **Both Referees#1 and #3 felt that the introduction wrongly emphasises the multi-decadal 'warming hole' over the exceptional freshening and interannual to decadal variability. We have edited *Section 1 Introduction* to address this, removing the reference to the multidecadal warming hole to focus on the interannual-decadal exceptional freshening event. We keep the overall structure of the introduction: wider context of TS variability in the ESPNA – details of the specific freshening event – aims and procedures of our study.**

Throughout, it's unclear to me exactly how the proposed mechanism agree/disagree/link to previous proposed mechanisms in literature. Formulations such as l.575 'we find neither of these to be fully convincing' make it sound like you discard all previous work, but surely your mechanism is not entirely independent of what's been proposed earlier?

> **We include more discussion on how our proposed mechanism relates to previous proposed mechanisms in *Section 7 Conclusions* as well as at various points in *Section 5*. Although it was not our intention to discard other work, we agree that we need to be more careful with our language and have modified this, while recognising that the reason**

> for embarking on this study was the idea that the previously published work did not offer a complete explanation of the freshening event.

l.314-316, l.322-324 and Fig. 8: You do see signs of an eastward shifted subpolar front, but you seem quite dismissive of horizontal redistribution in the conclusion (l.600-601). In addition to Holliday et al. 2020 and Desbruyres et al. 2021, Kenigson et al. 2020 and Asbjørnsen et al. 2021 are relevant for such horizontal shifts. Asbjørnsen et al. 2021 is also quite similar in terms of the particle tracking approach.

> **We think this is a fair point and substantially revise *Section 1: Introduction*, *Section 5.1 and Section 7: Conclusions* accordingly.**

> **This passage (l.600-601) was intended to be quite speculative, postulating that while horizontal redistribution is clearly occurring (see figure 1) it probably isn't the whole story and the mechanism identified here provides a potential route to additional vertical redistribution of heat and salt.**

> **An interesting additional perspective on both this question and the one above is provided in a recent manuscript of Volkov et al. (Ocean Science Discussions, *preprint*, https://egusphere.copernicus.org/preprints/2022/egusphere-2022-354/). They show both reduced heat loss associated with anomalous positive SSH anomalies in the Labrador Sea (associated weaker western SPG and deeper isopycnals), and that widespread cooling and freshening in the eastern SPG must be predominantly advective. Here we link those two processes.**

Are you seeing the subpolar trend reversal in VIKING20X as documented in Desbruyres et al. 2021?

> **We possibly begin to see some reversal of the subpolar trend (see e.g. Figure 7). It looks like this may be later than in Desbruyres et al., but our section is at the northern edge of the area they consider. We have not added discussion of this more recent period as outside the scope of the current work.**

Fig. 8a-b: Eastern and western parts look quite different. Interesting that OSNAP-E-37W-500m total (Fig. 5a) is dominated by what's happening at 37W-21W. A point that should be noted? The fractional evolution seen for the eastern part is more consistent with what is seen for the Nordic Seas inflow in Asbjørnsen et al. 2021.

> **We note this in the text (*Section 5.1, paragraph 4*). The dominance of western segment over the eastern segment is due to the larger volumes transported through the western segment, and may also be change dominating visually over no change. Asbjørnsen et al. (2021) consider the flow across the GSR to the Nordic Sea, the fractional evolution for this is probably more consistent with the eastern part because transport through GSR is dominated by Rockall Trough/Rockall-Hatton Bank transports. Iceland Basin and Irminger Sea northward transports, while larger, are more likely to recirculate and be transformed to deeper waters south of the GSR.**

l.358-360 and onwards: In any sort of AMOC change discussion, it needs to be clear that there is no consensus on whether the AMOC actually has systematically weakened over the 20[th] century until today (e.g., debate summarized in Jackson et al. 2022, Latif et al. 2022).

**Agreed. We mention this in *Section 5.2, paragraph 1.* We should also point out that long-term AMOC changes are not actually relevant to the sub-decadal scale changes we consider.**

l.445-446: Labrador Coastal Current and 'the other' – are both these branches what's typically called the Labrador Current? I also don't get the LC-Arctic and LC-Atlantic distinction (l.411-412).

**l.445-446: Yes.**

**l.411-412 The -Arctic and -Atlantic suffices in LC-Arctic and LC-Atlantic refer to the immediate upstream source of water in the Labrador Current. LC-Arctic has come south through the Davis Strait. LC-Atlantic water has predominantly come into the Labrador Sea from the Irminger Basin in the West Greenland Current. On reflection we concentrate on our source-based categories and introduce the LC-Arctic and LC-Atlantic terminology only to orient the current work in the context of that which has gone before. Various updates made to *Section 5.3.3 and the Conclusions.***

You make a convincing case for the freshening/cooling seen at OSNAP-E is due to a higher fraction of Lab. Sea water (Fig. 5, Fig. 7, Table 1) related to longer residence times in the SPG (Fig. 10a). I don't fully get why more SPG recirculation necessarily must be due to reduced heat loss and deepening isopycnals (l.435-441). Do you reference studies showing this mechanism for the SPG anywhere? Perhaps this point needs to be repeated in the conclusions.

**L.435-441: As worded, this paragraph is probably out of place, anticipating results in section 6. We have reworded it. The idea is that upper water crossing northwards across OSNAP-E does one of three things: crosses the GSR (where it is made denser and returns in deeper layers): is made denser within the SPG sinking to intermediate and deep layers; or recirculates in the SPG. To increase the proportion recirculating, one of the others needs to reduce. The most likely candidate is reduced conversion to denser water in the SPG as flows across GSR appear pretty constant from observations. *We make changes to Section 5.3.2, final paragraph and Section 7:Conclusions, paragraph 10.***

Lagrangian analysis:

Are particles released over the full OSNAP-East line as said in l.75-76 or only over the part of OSNAP-East shown in Figure 3? Are you analysing releases over the upper 1000m as said in l. 76 or the upper 500m as stated in l.111. I think the information in l.111-116 needs to come earlier to avoid conflicting messages.

***Section 2.2, paragraphs 1 and 6. Table 1.* Agreed. We clarify. We did release more particles than we used in the analysis. The reader does not need to know this though. Particularly those particles west of 37W. We present some headline results from the particles down to 1000 in response to comments by other referees (Table 1), so will keep the information about the particles released down to 1000 m.**

l.84-86: A bit unclear. How about explaining it as: 'Each particle represents a volume transport of 0.001797 Sv. The number of particles released along the 2-dimensional OSNAP-E section is scaled with the model velocity normal to the section at the release time.'

***Section 2.2, para 3.* We have modified this description in line with this suggestion to make it clearer.**

l.87-89: What is the reasoning for choosing Parcels and not a tool like Ariane where streamlines are computed analytically?

> ***No changes.*** **We don't believe the choice of tool affects the results and Parcels was chosen for pragmatic and operational reasons. Published experiments with Ariane and Parcels show very similar behaviour with appropriate choice of parameters (timesteps). E.g. Schmidt et al 2021.**

l.89-95: I don't quite get the type of errors discussed here (Errors induced by temporal resolution? Errors related to lacking diffusion? Errors related to particle release number? Errors related to assumption of stationarity?). What are the 32 subsets?

> ***Section 2.2, paragraph 4.*** **We attempt to clarify this description further in the text. The errors we quantify are those due to using finite particle numbers. As we increase particle numbers each metric we discuss will converge towards a 'correct' value. Using random release positions allows us to split the particle set into a number of subsets (32 here), each of which is still random but with each particle representing a (32 times) larger transport. The spread of the estimates from these subsets gives us an idea that we have released enough particles for these sampling errors to be small compared to the signal we are examining. We do not attempt to quantify errors due to lacking diffusion in the tracking, model biases, assumption of stationarity. We include mention of these error sources in a wider 'disclaimer' requested by Referee #1.**

Gulf Stream and Lab. Sea are defined as the two 'upstream origins' for OSNAP-E water (l.96). Then you subdivide Lab. Sea 'origin' by 'upstream origin' again (l.102): Hudson Bay, Davis Strait. Greenland Sea, Denmark Strait, SPG. Later in the manuscript the word 'source' is frequently used. I would be very clear with the definitions and use them consistently throughout. Perhaps Lab. Sea and Gulf Stream is your 'upstream sources' while the Labrador source region is subdivided by four 'origin regions'?

> ***Section 2.2, paragraph 5 and onwards.*** **We make these definitions and uses consistent, using 'source' for the Gulf Stream/Lab Sea/other division, and 'upstream source' for the Hudson Bay, Davis Strait. Greenland Sea, Denmark Strait, SPG . We try to use 'source' for the region and 'water of xxxx origin' or 'xxxx origin water' for water or particles which came from 'xxxx' source region. We hope this is now clearer.**

Figure 3: I'm confused by the lines on the map. Lines over land should be gone? Are particles crossing the orange line cutting across the Lab. Sea defined as having Lab. Sea origin? Is the north-south orange line the Loop path/Slope Sea pathway definition? I get the east-west green line is the Gulf Stream definition, but what about the north-south thin green line? What about showing definitions for Davis, Hudson, Greenland Sea origins in panels f-h, respectively?

> ***Figure 3.*** **We have hopefully cleared this up with the production of a revised figure 3.**

l.209-212: Such fractions will depend on the depths evaluated over and the source definition used. In Koul et al. 2020, particles are initialised in the upper 100m only. In Asbjørnsen et al. 2021, 26% of the Iceland-Scotland Ridge inflow (upper 1000m or so) has a subpolar or Arctic origin (Davis, Hudson, Denmark straits, or circulation in the SPG), with 42% of the surface inflow being water from Davis Strait and Hudson Bay.

*Section 4.1, para 1.* **We expand this discussion to include Asbjørnsen et al. 2021 numbers, and include ranges on our fractions from the different initial particle distributions (500 m, 1000m, AMOC upper limb).**

Minor comments / technical corrections:

1. l.80: reference in parenthesis or write 'shown in Biastoch et al. 2021'.

    *New line 82.* **Changed.**

2. Fig. 1 and Fig. 2 is not referenced in the text until page 9. Either reference earlier or rethink figure-order.

    *No change.* **These figures are mentioned first in the introduction, so we think they are in the correct position.**

3. l.107-110: Would shorten this point. Doesn't need to be its own paragraph.

    *New lines 113-115.* **Changed.**

4. Section 2.4: Unless you want to expand on OSNAP measurements and how the EN4 product is produced, I would delete this section and move the sentence somewhere suitable.

    *Section 2.4.* **This section remains as it is now longer with the inclusion of details of the Labrador Sea data used in the revised model-data comparison in section 6.**

5. Section 2.5: Not really necessary information – I recommend deleting the section and put the information about code availability in the data statement at the end.

    *New lines 676+* **(AF) think this is important information for openness and reproducibility. We move it to the code availability statement.**

6. l.216-217: Stated already in section 2.2, not necessary to repeat.

    *New lines 225* **Deleted**

7. Starting sentences with 'So' or 'But' (sometimes) making them not 'full sentences' with a subject, verb, and an object: e.g., l.220, l.247, l.255, l.269, l.427, l.465, l.519, l.546.

    *Various places.* **We've checked for and removed these style and English usage issues**

8. l.220-223: No need to repeat the details here.

    *New lines 232* **Deleted**

9. l.294-305: I would cut this section and save such summarizing statements for the conclusion. If you are worried some of the interesting results might be 'lost' because the manuscript is quite dense, you could even do a bullet point list summarizing the main results in the conclusion.

    *No change.* **We think this section is useful for readability, summarizing results from quite a dense section before moving on to new sections of results/discussion which draw heavily on section 4. We have decided to leave it in place.**

10. l.337-340: check punctuation and parentheses.

11. l.341-344, l.411:413: check parentheses.

**Done**

12. Section 5.3 title: title sounds like you again will discuss what was addressed in 5.1. I would find an alternative pointing to the Labrador Sea focus.

    *Titles of sections 5.1, 5.2 and 5.3* **have been changed to better reflect the content.**

13. l.414-416: ref. Fig. 10?

    *New line 450.* **Added Figure reference.**

14. Fig. 10 and l.414-419: What depth is the 'light upper layers' - are you still looking at 0-500m?

    *New line 454-5.* **These are still the particles we have tracked backwards from the top 0-500m at OSNAP. Further upstream they are not strictly confined to the surface 500m. As the subpolar region is predominantly an area of sinking, particles launched in the upper layers and tracked backwards will generally stay in the upper layers. We make this clearer in the text.**

15. l.528-532: Too long sentence – difficult to decipher.

    **Section 6.3 has been rewritten with different observational data. See the response to Referee #1, general comment 3 for more details. So this comment has been superseded.**

16. Legends Fig. 7, Fig. 8, Fig. 10 looks quite messy. I would display text as a structured column. Fig. 8 could have one common legend on the side.

    *No change.* **I (AF) disagree here. I find it difficult switching between viewing the graph and the legends to remind myself which line is which. I find labels by the lines on the plots are much easier.**

Mentioned literature:

Kenigson et al. 2020 – 10.1175/JPO-D-20-0071.1

Asbjørnsen et al. 2021 –  10.1175/JCLI-D-20-0917.1

Desbruyres et al. 2021 - 10.1038/s43247-021-00120-y

Jackson et al. 2022 -  10.1038/s43017-022-00263-2

Latif et al. 2022 - 10.1038/s41558-022-01342-4

**Anonymous Referee #2**

**We thank Referee #2 for their constructive comments.**

This article sheds new light on the main drivers behind the cooling and freshening recently experienced in the upper eastern subpolar North Atlantic. To this end, the authors analyze the

outputs of a historical hindcast (years 1980 to 2019) performed with the eddy-rich ocean–sea-ice model VIKING20X, using a variety of techniques and diagnostics, from lagrangian particle tracking, to water mass transformation analyses.

The article is well-timed and well written, the methodology applied is sound, figures are clear, and many of the results will be of high interest to the climate community at large, and to anyone with specific interests in the recent changes experienced in the North Atlantic and its surroundings.

I recommend a minor revision of the manuscript and enclose a list of comments that the authors would need to address to render the article suitable for publication in *Ocean Science*.

General Comments:

- Agreement of the VIKING20X-JRA hindcast with observations is mentioned several times throughout the text, but it is mostly derived from visual comparisons, which can be misleading. In some cases the agreement is clear, but in other cases is much less evident (see comments #4, #16 and #18 further down). Supporting these statements with some specific metrics, like linear correlations between the hindcast and the observations, will help to determine more precisely to what extent they agree with each other.

*This is a wide-ranging comment and we have made changes to Figures 1 and 2, 14, Sections 3, 6 and 7 to address it.* **Model verification has been done extensively in Biastoch et al. (2021) which we refer to frequently throughout the manuscript. We do not want to repeat that verification here. The purpose of the initial comparison between the model and EN4 analysis (figures 1 and 2) is to show that the VIKING20X model produces a freshening and cooling signal analogous to, if not exactly the same as, that in observations, to support the use of the model to investigate this phenomenon. We feel that the visual comparisons presented are sufficient for this purpose and don't believe that further quantitative metrics comparing hindcast and observations are necessary. We try to make the limitations of the use of an imperfect model, and the fact that what we present is a model-based hypothesis rather than objective truth, clearer in the discussion, as requested in a similar point made by Referee #3.**

**While the EN4 analysis is of sufficient quality and resolution for the initial comparison aimed at supporting the use of VIKING20X in this study, we find it to be of insufficient quality for the more detailed comparison required in the Labrador Sea in Section 6.3, resulting in the, fully justified, criticisms of this comparison by all Referees. Here, we completely rewrite the comparison, using observational data (rather than the EN4 analyses) in the Labrador Sea. The new dataset was provided by Igor Yashayaev, tailored specifically to our Labrador Current outflow region and based on the latest version of the data presented here: https://waves-vagues.dfo-mpo.gc.ca/Library/40974698.pdf. The comparison between model and observations here is much closer using this improved dataset than with EN4, with the observational data clearly showing deepening isopycnals. While this new dataset and EN4 analysis contain much of the same underlying data (Argo floats and moorings) the improved quality control, analysis focussed specifically on our area and variables of interest, and removal of relaxation to climatology in the absence of data, make the switch to this new dataset appropriate. While these remain purely visual, qualitative comparisons, we hope that because the similarities between model and observations are now clearer this will allay the Referees' concerns.**

- Given the comprehensive list of processes and mechanisms that are analyzed in the paper, which in many cases are interconnected, it would be very useful to include at the end of the

article a schematic figure summarizing the chain of events that give rise to the freshening and cooling of the eastern subpolar North Atlantic, as supported by the model analysis.

***Figure 15 and conclusions.*** **This is a nice suggestion and we have added an appropriate diagram as Figure 15, referred to from the Conclusions section.**

- Several figures from other articles are cited throughout the text, urging the readers to keep jumping from one article to another, and thus hindering the overall readability of the article. Some of those figures include indices that could be easily incorporated in others figures of this manuscript (see comments #6, #10 and #11), making intercomparison between those and the VIKING20X indices much more straightforward. I strongly recommend the authors to include them.

***Figures 5 and 9.*** **We have included subpolar gyre index in figure 5 and RAPID AMOC transport in Figure 9.**

Specific Comments:

1. [Lines 18-19] This sentence seems incomplete and inaccurate. It should (1) mention that there is a cooling on the eastern subpolar North Atlantic, not a surface warming anomaly, and (2) it should also specify that this is simulated in response to a "weakening" of the ocean circulation. Also, I recommend the authors to avoid the use of the term "predict" in the sentence, as "predictions" generally refer to historical simulations initialized from observations to phase the model with the observed internal variability. However, the warming hole is a feature that consistently appears in uninitialized historical simulations, and is therefore deemed to be mostly externally forced.

***The opening sentence has been removed.*** **In line with suggestions from the other reviewers we have removed the unnecessary and somewhat confusing references to the multi-decadal 'warming hole' and focussed the introduction on the short-term exceptional freshening and cooling event. We keep the format of the introduction: wider context of TS variability in the ESPNA – details of the specific freshening event – aims and procedures of our study.**

2. [Line 108] It is unclear which "common technique" it refers to.

***New line 114.*** **We have removed the reference to 'common technique'**

3. [Figure 4b, caption] Is it "number of days" or "number of years"?

***Figure 4 caption.*** **Years. The caption is corrected.**

4. [Lines 177-181, Figure 2] While the simulated upper temperature variability in the hindcast shows a good agreement with the observed timeseries from both OSNAP and EN4 products, this is not the case for the simulated salinity. This is particularly clear for EN4, which can be compared for a substantially longer period, showing large discrepancies in terms of both the high and the low-frequency variability. Indeed, a close inspection to Figure 2 reveals that the differences between EN4 and VIKING20X-JRA are not stationary in time, and reflect more than the systematic mean state bias stated in the text. It is true that salinity observations are quite scarce and therefore objective analyses like EN4 are subject to large uncertainties, but it remains to be checked whether the simulated variability is within the range of the observed uncertainty.

*Figure 2, Section 3 text*. **We have replaced Figure 2 with a revised comparison and adjusted the text accordingly. We now compare x-z area averages of temperature and salinity rather than depth-averages and include EN4 confidence intervals. We also start the comparison at year 2004 when comprehensive coverage by Argo floats becomes available. Prior to the early 2000s salinity data were very sparse and EN4 salinity analyses largely reflect climatology. EN4 is a lot smoother both in space (see Fig. 1) and time. Even when averaging in space (Fig. 2) or time (Fig 1) there will always be mesoscale structure in VIKING20X output that shows as variability exceeding that of EN4. As stated in response to the general comments above, we believe that these comparisons now demonstrate that the model shows sufficient skill in reproducing the particular freshening event to justify its use here. We add consideration of the implication of model skill on the conclusions we have about the mechanisms involved in the discussion.**

5. [Lines 311-313] What do you mean by "contrasting evolution of transit times"? Are their associated histograms (like in Figure 4) substantially different? And in which way?

*New lines 328+* **This point was not about the histograms in Figure 4, but about the time series in Figure 8. Particles crossing OSNAP-E in the west showed marked lengthening of transit times (and longer track lengths and slower mean speeds) in the 2000s, contrasting with particles crossing further east which showed much steadier speeds and transit times. This will be clarified in the text.**

6. [Lines 334-340] Instead of referring the reader to Figure 10 in Biastoch et al 2021, it would be preferable if the authors included the SPG index in a subpanel of Figure 9, where it can be directly compared with the AMOC indices.

*Figure 5, and line 363.* **We include the SPG index from Biastoch et al 2021, but in Figure 5 rather than Figure 9. The AMOC from RAPID observations is included in Figure 9.**

7. [Lines 339-340] The two parentheses referring to Figures 5 and 7 need to be closed.

**Done**

8. [Lines 363-364, Figure 9] Why did you choose to plot the AMOC at 29°N and not at the same latitude of the RAPID array (i.e. 26°N)? It would be indeed very interesting to include a direct comparison of the hindcast with the RAPID data, to learn more about the model realism in representing dynamical aspects, like the North Atlantic ocean circulation.

*New lines 392+* **This direct comparison between VIKING20X AMOC and the RAPID array is in Biastoch et al. (2021), referred to in the text, and does not need to be repeated here. On the choice of latitude 29°N for the calculations, we will add brief explanation to the text, and include more details here. There is very little, if any, difference between AMOC at 26°N and 29°N and the choice was made entirely for ease of calculation.**

**We chose 33°N for the Lagrangian analysis because it was the furthest north it could be (so keeping tracking times as short as possible) and be confident that particles caught crossing there were from the western STG rather than recirculating in the inter-gyre region for example.**

**However, for the Eulerian analysis we chose 29°N because the flow there is really simple. It lies at about the central latitude of the STG. As it is clear of the Antilles (unlike 26°N), practically all the northward upper layer flow sits on the wide 1000m deep shelf. Further offshore, almost all the DWBC is above the slope at the edge of this shelf, clearly separate in**

the horizontal from the northward WBC. Then further east still is the broad southward flow in the eastern STG. All three of these are clearly separate in the horizontal, making calculation of STG strength, AMOC, DWBC transport and STG to SPG surface flow all straightforward. Matching the Eulerian section with the Lagrangian at 33°N would make these calculations more complex, with some northward WBC currents overlying the southward DWBC, and many more eddies and recirculations east of the Gulf Stream. Matching with Rapid at 26°N was attractive, but the comparison is not our focus, is included in Biastoch et al. (2021), and again the flow is more complex at 26°N than at 29°N, with the WBC divided between the Florida Straight and the Antilles current, with the latter overlying the DWBC.

9. [Lines 366-368] I would not say that the AMOC is responsible for a reduction in Gulfstream transport. The Gulfstream does not respond to the AMOC, it is a component of the AMOC and as such contributes to its variability, not the other way around.

*New lines 400-402.* **We have adjusted the wording. Variability in the GS can be associated with a combination of AMOC and STG variability, as can be seen in Fig 9. We do not mean to claim that AMOC is responsible for a reduction in Gulf Stream transport, but rather for the reduction in transport from the GS source to OSNAP$_E$.   That is upper layer flow northwards from suptropical to subpolar gyres that we measure is associated with the AMOC weakening. We reword and change 'responsible for' to 'associated with' to remove any claim on causation.**

10. [Lines 371-372] This can be directly shown to the reader if, as suggested in Comment #6, the SPG index is included in Figure 9.

*Figure 5.* **We include the SPG index, in Figure 5 rather than Figure 9 as this felt more appropriate.**

11. [Lines 378-380] Here you compare again with an observed index from another article that could be easily included in this one (in Figure 9).

*Figure 9.* **We include the RAPID AMOC transports in Figure 9.**

12. [Line 386] I do not find this summary statement fully justified. There has been no specific analysis in section 5.2 to rule out the AMOC weakening as a main contributor to the freshening of the eastern subpolar North Atlantic.

*New lines 422+.* **Agreed, hopefully this is now clearer, and the possible contribution from the Gulf Stream source is better represented. The analysis showing that reduced transport from the Gulf Stream source was not the majority contributor to the freshening was in section 4.3, but the contribution was still estimated at a sizeable minority of 30%. Section 5.2 confirms that this reduced contribution from the Gulf stream source is reduced AMOC.**

13. [Line 394] The parenthesis needs to be closed.

*Done*

14. [Lines 520-521] It is important to specify that the doubling of the annual mean depth of those isopycnals only occurs in VIKING20X-JRA. In EN4, the increase in the depth of the isopycnals is very subtle and only discernible for σo > 27.65 kg/m3.

15. [Lines 526-528] Figure 14 a-c → Figure 14c (as this is the only panel showing the isopycnals)

16. [Lines 537-539] From Figure 14 it is not possible to say if the model realistically represents the variability in salinity and density. Indeed, as mentioned in Comment #14, the strong

deepening of the isopycnals in VIKING20X-JRA (one of the major reasons behind the freshening in the eastern subpolar North Atlantic identified in the paper) is not seen in EN4, and in particular in the upper layers.

***Section 6.3 and Figure 14.*** **We have rewritten Section 6.3 based on comparison with a better quality-controlled and more carefully analysed Labrador Sea observational dataset and a new Figure 14. This new comparison shows the strong deepening of isopycnals in the relevant depth range in both data and model.**

17. [Lines 544-545] Can you explain why it would be counter-intuitive?

***New line 584.*** **This was simply that we suggest *warming* in the surface Labrador Sea leads, purely by internal ocean advection, to *cooling* in the eastern subpolar gyre. We have removed the phrase.**

18. [Lines 589-593] Not all the model results are fully supported by observational data, in particular the deepening of the isopycnals in the Labrador Sea. Furthermore, the evolution of the Labrador Sea vertical structure only compares well with the EN4 data (Figure 14) for temperature. The agreement with the observed salinity and density is much more limited. It would be worth discussing whether, and if yes, in which sense, this poor agreement affects the main findings derived from the VIKING20X-JRA analysis.

***Section 7, conclusions. New lines 633-643, and 664+.*** **Having revised the model-observation comparisons we feel that the original statements are now well supported, so we leave the main message of this paragraph unchanged. We include a new 'disclaimer' paragraph in the conclusions highlighting possible limitations of the study.**

**Anonymous Referee #3**

**We would like to thank Referee #3 for their constructive comments. In common with other Referees, Referee #3's major concerns are with the model-observation comparisons at the start – supporting the use of VIKING20X – and at the end – in support of our conclusions about a proposed mechanism. We substantially revise both these sections and provide drafts of revised, more relevant, Figures which better support our conclusions.**

In this study, the authors investigated mechanisms of the exceptional freshening event in the eastern subpolar North Atlantic using a high-resolution VIKING20X model run. The authors conducted a thorough study starting with a Lagrangian tracking analysis that leads step by step to the conclusion that the freshening event in the eastern subpolar gyre is due to reduced heat loss in the western subpolar gyre. Overall, I find the manuscript logically organized with convincing conclusions. However, I believe several general concerns need to be addressed before the manuscript can be accepted for publication.

General comments:

1. This study is based on the high-resolution VIKING20X-JRA-Short hindcast simulation, which is able to capture the great freshening and cooling event in 2014-2016. The pattern and timing

of the freshening and cooling from the model is very consistent with those derived from EN4. However, the magnitude of the freshening and cooling in the model is substantially larger than in EN4. Since the hindcast is a free run without data simulation and bias correction, the authors argue that the model serves as a dynamically consistent tool to examine the freshening event.

*Figure 1,2 and Section 3 have been substantially revised.* **The original mapped comparisons between VIKING20X and EN4 (Figure 1) exaggerated the differences in magnitude of the freshening and cooling. This was due to the use of a long reference period during which model and EN4 temperatures and salinities were different. These differences in the means fed into the anomaly calculations resulting in the model freshening and cooling signal appearing have a notably magnitude than the signal in EN4. This exaggeration could be seen by examining the time series (Figure 2) where freshening and cooling between 2012 and 2016 were of much more comparable magnitudes. To remove this misconception we have removed the unrealistic basin-scale long-term trend from the model before producing new figures 1 and 2 (Figure 1.1 and Figure 2.1 here) and focussed more closely on the exceptional freshening and cooling signal by showing anomalies from a shorter period during which EN4 and model were temperature and salinity fields were closer. Figure 1.1 and 2.1. This may allay some of the concerns expressed in the first comment below.**

This leads to my first concern: given the large simulated bias in the magnitude of the freshening/cooling event, the model could be overly sensitive to a certain mechanism (e.g., heat loss in the Labrador Sea) that contribute to the freshening, while it may underestimate other mechanisms (e.g., AMOC). The authors point out that warm and salty bias prior to the 2014-2016 event is a common feature of hindcast simulations. However, this is not a valid argument that we shall expect a stronger freshening in the model than in observation. Can the authors explicitly explain the reason for this much stronger freshening in the model? Do other realizations of the VIKING20X hindcast simulations show a similar freshening event? If the answer is yes, can the same mechanism explain the freshening? In any case, a clearly stated disclaimer is needed in the discussion to remind readers that compared to observations, model bias both prior to and during the freshening event is strong. The proposed mechanism may be subject to model bias.

*Section 3, Figs 1 and 2, Conclusions, particularly lines 666+.* **The bias in magnitude of the freshening and cooling event is not as large as it appeared in the original version of Figure 1, but we accept the point that the model could be over-sensitive to some mechanisms and under-sensitive to others. This is part of the reason why we present our conclusions as a model-based hypothesis. It is incorrect to characterize the model signal as 'much stronger freshening' – a misconception which was entirely our fault due to the presentation of the original Figure 1. We cannot explain the differences in detail between model and observed signals, and feel this is beyond the scope of the current paper in which we attempt a coherent explanation of the mechanisms behind the overall freshening rather than particular details. We have not examined the freshening event in other realizations of the VIKING20X hindcast and, due to the complexity of particle tracking and the time and resources involved, it is not possible to repeat the particle tracking in other model runs.  We feel that the new, more representative comparison between the freshening event in the model and the EN4 analysis supports the use of this hindcast without examination of other VIKING20X hindcasts, and as suggested, we address the points about the possible influence of model bias on the results in a clearly-stated disclaimer.**

2. In section 5.2, the role of the AMOC in driving the freshening is discussed. It is concluded that the weakening Gulf Stream source, determined via particle tracking, is associated with the AMOC in the subtropics and is not related to the subtropical gyre circulation. What I find missing here is the AMOC in the subpolar north Atlantic. What is the role of the subpolar AMOC in the great freshening event? Does the subpolar AMOC also weakens around a similar timing in VIKING20X? How is the magnitude and time of the weakening (or maybe strengthening) of the subpolar AMOC in the model compared to the subtropical AMOC? The authors have studied the great freshening event in the subpolar North Atlantic, used OSNAP East section as the termination of the Lagrangian tracking method, and determined that the Labrador Sea heat loss plays a key role in driving the freshening event. However, the authors have avoided investigating the AMOC in the subpolar North Atlantic.

*No change.* **While this is an interesting question, which we hadn't previously considered, we do not believe it offers additional insight into the causes of the freshening and cooling event so we do not include it in the paper. Our reasoning is as follows:**

**Biastoch et al. 2021 (referenced in the paper) do a detailed analysis of subpolar and subtropical AMOC in VIKING20X with comparisons with RAPID and OSNAP data, so we do not include that here. We also don't find the subpolar AMOC index to be useful tool in diagnosing the causes of the freshening (at least not as useful as the subtropical AMOC index which can be considered a measure of the volume of water exchanged between subpolar and subtropical regions). This is because subpolar-gyre-scale mechanisms with the potential to cause the observed freshening and cooling can be effectively neutral with respect to subpolar AMOC. However, it is interesting to consider how the proposed mechanisms interact with subpolar AMOC (considered in density space). Both the main mechanisms we consider involve the upper limb of AMOC. The reduced flow from the Gulf Stream source, associated with reduced subtropical AMOC, will just feed through into reduced subpolar AMOC as the transformation to denser water is almost entirely north of the OSNAP line. The subpolar-gyre scale mechanism is more complex. The increased southward flow of lighter water from the Labrador Sea, could be effectively neutral on subpolar AMOC in the longer term as the lighter water leaving the Labrador Sea recrosses northwards. The time difference between these crossings should contribute to interannual changes in subtropical AMOC, but these are difficult to detect among the other sources of upper limb subpolar AMOC variability. However, the mechanism identified behind the increased outflow of lighter water involves reduced transformation of lighter AMOC upper limb water to denser lower limb water in the Labrador Sea north of the OSNAP line. This should directly reduce AMOC measured at both subpolar and subtropical latitudes, with some lag.**

3. I do not find Section 6.3 convincing. First, I do not see how the modeled isopycnal depths "agree closely" with observation. The model has substantially shallower isopycnals, particularly at large depths. This means that the model has a substantially higher density in the Labrador Sea. However, I cannot understand how the model can have fresher and warmer water and at the same time higher density throughout the water column.

*Section 6.3, Figure 14.* **Problems with this section were highlighted by two of the Referees, many of the concerns were due to problems of comparison of the model with EN4 analyses; the different resolutions, the use of climatology to fill gaps in EN4 salinity in particular. We overhaul this section completely with comparison to a more appropriate dataset specific to the Labrador Sea outflow region over the modelled period, and a new**

**Figure 14 (included here as Figure 14.1). We are now careful to distinguish between mean misfits (models typically cannot match sigma up to the second decimal place) and changes/trends. This comparison is much stronger and we feel the isopycnal depth changes – the fundamental variable in our argument -- do 'agree closely' with this more relevant observational dataset. The new dataset was provided by Igor Yashayaev, based on the latest version of the data presented here: https://waves-vagues.dfo-mpo.gc.ca/Library/40974698.pdf. We add a brief description of this dataset in the methods section.**

**For completeness, and as the discussion paper remains available, we thank Reviewer #3 for spotting the mistake here. I (Alan Fox) have mixed Absolute Salinity (for EN4) and Practical Salinity (for VIKING20X) in the original Figure 14, explaining how the model appears fresher and warmer and at the same time denser.**

Minor comments:

1.  I do not think it is necessary to start the introduction with the "warming hole". It might be more straightforward if you directly start with text describing the recent "freshening and cooling" event.

*Various changes to Section 1:Introduction and final lines of Conclusions.* **Two Referees made similar comments, and on reflection we agree. We have removed the reference to the warming hole, but we keep the format of broad-scale context – specific event studied – aims and procedures.**

2.  In Figure 2d, why is there direct-path water crossing 60°W.

*Figure 3.* **I think this refers to Figure 3d. It was explained in the caption as water which has made more than one complete loop of the subpolar gyre, and we allocate paths on the first loop. However, a new version of the figure is included to remove the confusion. (attached here as Figure 3.1)**

3.  Section 5 need to be reorganized. Both section 5.1 and 5.3 are subpolar-gyre-related mechanisms. And what does it mean by basin-scale in section 5.2? Gyre circulation is also basin-scale.

*Section 5.1,5.2,5.3.* **We leave the sections here ordered as before – we tried other ways of organising these and this felt the most natural. However, we clear up the section titles, basing the titles of sections 5.2 and 5.3 on the origin of the water being discussed. We were using 'basin-scale' to signify whole North Atlantic (or even whole Atlantic) scale processes, and 'gyre-scale' for processes centred on the subpolar region.**

4.  Line 364, how is "subtropical gyre recirculation" defined?

*New lines 398.* **We have defined this, in combination with an explanation of our choice of 29°N requested by another referee. We use subtropical gyre recirculation to refer to that part of the northward Gulf Stream flow which returns southward in the broad upper ocean interior flow in the subtropical gyre, as against that part which continues northward to subpolar latitudes.**

5. Line 377, I do not think there is a consensus on whether the AMOC has declined since the 1990s. Models and proxies suggest that AMOC has declined (e.g., Rahmstorf et al., 2015; Ceasar et al., 2018, 2021), while observation-based reconstructions have not found a significant AMOC decline (Fu et al., 2020; Worthington et al., 2021; Caínzos et al., 2022).

*New lines 399-402, 384-390.* **We change this section to refer to the reduction in AMOC strength observed in RAPID from before 2009 to after 2009 rather than an AMOC 'decline' and include new text and references on the lack of consensus around a decline.**

6. Lines 459, please specify the density of the "lighter" waters.

**Done.**

7. Lines 461, it is confusing to call waters lighter than 27.50 kg m$^{-3}$ as the "lightest" waters.

*New line 499.* **OK. We clarify this.**

8. In Section 6.2, it is concluded that due to reduced heat loss over the Labrador basin, transformation from lighter to denser water mass is weakened. Therefore, the steady inflow in the upper layer (<27.65 kg m$^{-3}$) must be balanced by an enhanced outflow also in the upper layer. Does this indicate that the overturning in the Labrador Sea weakens, while gyre circulation in the Labrador sea strengthens? This leads back to Section 5.1, lines 335-340, where it is found that the SPG is not responsible for the freshening. How would the authors reconcile the discrepancy here?

*Section 6.2. Mainly new lines 556-568.* **We clarify this to remove what we think is possible confusion rather than a discrepancy. The reduced transformation does indicate reduced overturning at this density in the Labrador Sea, but it does not necessarily follow that gyre circulation strengthens. We find increasing lighter water outflow at a time of weak, and constant or further weakening SPG strength (the appropriate SPG index will be included in Figure 5). Rather than being associated with increased gyre circulation (which is mostly barotropic) the increased upper layer outflow is associated with reduced lower layer outflow.**

9. 14 needs to be reorganized. Fig. 14(d,e,f) is cited before Fig. 14(a,b,c).

*Figure 14.* **This figure has been superseded by replacement Fig 14 and rewritten section 6.3.**

10. Line 536, salinity issue does not make temperature comparison reliable, the sentence needs rephrasing.

*Figure 14 and Section 6.3.* **Superseded by the full overhaul of the model-observation intercomparison as detailed under the general comments above.**

Reference:

Rahmstorf, S., et al. (2015). Exceptional twentieth-century slowdown in Atlantic Ocean overturning circulation. Nature Climate Change, 5(5), 475–480

Caesar, L., et al. (2021). Current Atlantic meridional overturning circulation weakest in last millennium. Nature Geoscience, 14(3), 1–120

Caesar, L., et al. (2018). Observed fingerprint of a weakening Atlantic Ocean overturning circulation. Nature, 556(7700), 191–196

Fu, Y., et al. (2020). A stable Atlantic meridional overturning circulation in a changing North Atlantic ocean since the 1990s. Science Advances, 6(48), eabc7836

Worthington, E. L., et al. (2021). A 30-year reconstruction of the Atlantic meridional overturning circulation shows no decline. Ocean Science, 17(1), 285–299

Caínzos, V., et al. (2022). Thirty years of GOSHIP and WOCE data: Atlantic overturning of mass, heat, and freshwater transport. Geophysical Research Letters, 49, e2021GL096527

**References:**

**Schmidt, C., F. U. Schwarzkopf, S. Rühs, and A. Biastoch, 2021: Characteristics and robustness of Agulhas leakage estimates: an inter-comparison study of Lagrangian methods,** *Ocean Sci.,* **17, 1067–1080, https://doi.org/10.5194/os-17-1067-2021.**

---

## Referee Report (RR1)

The authors have done a thorough job at revising the manuscript. Overall readability is improved, and it is now much more clear how the results align/contrast themselves to previous work. I also appreciate the clarifications regarding e.g. particle sampling errors and the Labrador Current components. I would be happy to see the manuscript accepted (I only have a few very minor comments):

1. l. 78: 'while not excluding the southward upper ocean flow east of Greenland.' Phrasing not entirely accurate considering the East Greenland Current is not a part of the release section?

2. l.130: 'variously calculated'?

3. l.228: 26 Sv not 26 m.

4. l.572: check citation parenthesis.

5. L.574: 'a positive anomaly in the net surface heat flux into the Labrador Sea (i.e. reduced heat flux out of the Labrador Sea)'. Unclear if you don't define the direction of positive/negative heat flux. Perhaps just formulate as 'reduced surface heat loss' (which is used in the following section anyways).

6. This might be slightly beyond the scope, but in identifying reduced surface heat loss between 2000 and 2013 as the 'trigger' for the cooling/freshening in the following years, a natural question is what caused the reduced heat loss. If you look at the accumulated NAO index (over the 10 years prior) you will see a peak of positive NAO values around 2000 and a peak of negative NAO values around 2011-12 – meaning that for the years in-between there has been a tendency for more negative NAO states (typically accompanied by reduced heat loss over the SPNA). Perhaps you have some reflections on this (NAO, heat loss etc.) for the conclusions.

---

## Author Response (AR2)

**OS-2022-18: Author responses to Referees**

**Referee's comments are in plain text, authors' comments in response are in bold.**

**Anonymous Referee #1:**

**We thank Referee #1 for their comments, our responses to the technical corrections requested are detailed below.**

The authors have done a thorough job at revising the manuscript. Overall readability is improved, and it is now much more clear how the results align/contrast themselves to previous work. I also appreciate the clarifications regarding e.g. particle sampling errors and the Labrador Current components. I would be happy to see the manuscript accepted (I only have a few very minor comments):

1) l. 78: 'while not excluding the southward upper ocean flow east of Greenland.' Phrasing not entirely accurate considering the East Greenland Current is not a part of the release section?

**Agreed. This should have been 'while excluding', we have removed the 'not'.**

2) l.130: 'variously calculated'?

**Changed to 'calculated both as…. and by….' as there are just two distinct methods used to calculate these averages.**

3) l.228: 26 Sv not 26 m.

**Corrected.**

4) l.572: check citation parenthesis.

**Corrected.**

5) L.574: 'a positive anomaly in the net surface heat flux into the Labrador Sea (i.e. reduced heat flux out of the Labrador Sea)'. Unclear if you don't define the direction of positive/negative heat flux. Perhaps just formulate as 'reduced surface heat loss' (which is used in the following section anyways).

**We think the positive direction was defined as 'into' the Labrador Sea, however we have adopted the suggested simplification.**

6) This might be slightly beyond the scope, but in identifying reduced surface heat loss between 2000 and 2013 as the 'trigger' for the cooling/freshening in the following years, a natural question is what caused the reduced heat loss. If you look at the accumulated NAO index (over the 10 years prior) you will see a peak of positive NAO values around 2000 and a peak of negative NAO values

around 2011-12 – meaning that for the years in-between there has been a tendency for more negative NAO states (typically accompanied by reduced heat loss over the SPNA). Perhaps you have some reflections on this (NAO, heat loss etc.) for the conclusions.

**Lines 632+. We have added a couple of sentences in the Conclusions about the source of this reduced heat loss and the possible relationship between the cumulative atmospheric forcing, ocean heat content, isopycnal depths and SPNA freshening. We refer readers to the (already referenced) work of Yashayaev and Loder (2017) where these ideas are discussed in more detail.**

**Anonymous Referee #1:**

The authors have addressed all my concerns. I appreciate the authors' efforts on the modifications, especially for the comparison of isopycnal depth between the model and observation. I recommend to accept the manuscript as it is.

**We thank Referee #3 for their comments. The requested revision of the comparison of isopycnal depth between the model and observation allowed us to examine this in more detail we feel resulted in significant improvements and increased confidence in the presented results.**